# IndexMem: Learned KV-Cache Eviction with Latent Memory for Long-Context LLM Inference

Xintong Yang [* 1]  Hao Gu [* 1]  Binxing Xu [2]  Lujun Li [1]  Bei Liu [1]  Jiacheng Liu [1]  Qiyuan Zhu [1]
Yike Guo [† 1]  Sirui Han [† 1]

## Abstract

Large Language Models (LLMs) are increasingly expected to operate over long contexts, yet standard softmax attention incurs a KV cache that grows linearly with sequence length, quickly becoming the bottleneck for long context inference. A practical remedy is to evict less important KV entries; however, existing eviction policies are largely heuristic and struggle to capture the rich, input-dependent distribution of token importance. In this work, we introduce a **learnable indexer** that predicts KV importance, enabling more accurate retention of critical tokens. Meanwhile, naively evicting tokens permanently discards their information, leading to irreversible forgetting and degraded retrieval over long ranges. To address this, we propose a lightweight **latent memory module** that compresses evicted tokens into a compact, online-updated state and provides residual readouts to compensate for the attention contributions lost through KV eviction. Collectively, our method enables accurate long-context inference under a bounded KV budget, delivering consistent improvements on RULER (4K/16K) across Qwen, Mistral, and Llama models (up to 25 points under aggressive eviction), markedly more stable Needle-in-a-Haystack retrieval, and superior LongBench scores and compression curves compared to existing eviction policies.

## 1. Introduction

Large Language Models (LLMs) have exhibited remarkable prowess in long context understanding, powering sophisticated applications such as complex reasoning over mathematical (Veeraboina, 2023) and coding problems (Jain et al., 2024), as well as agentic workflows like making slides (Manus, 2024) which demands large scale multimodal input processing (Team et al., 2025; Georgiou, 2025). Leading models like Gemini3 (Pichai et al., 2025) now support context windows of up to 1-million tokens, a scale readily encountered when ingesting video streams or extensive codebases.

However, autoregressive generation poses a critical resource bottleneck for long context inference. To avoid the quadratic complexity of recomputing attention at each decoding step, modern inference engines leverage Key-Value (KV) caching to preserve intermediate states of previously processed tokens. While this strategy substantially reduces computational overhead, it displaces the bottleneck from computation to memory: the KV cache footprint scales linearly with both sequence length and batch size. For instance, maintaining a KV cache for a 1-million-token context in a 70-billion parameter model consumes approximately 320 GB of GPU memory (Grattafiori et al., 2024), far exceeding the capacity of most commodity accelerators. This "memory wall" constrains not only the maximum deployable context length but also exacerbates decoding latency, as massive KV data movement saturates memory bandwidth. Recent advances in KV cache compression, such as quantization (Liu et al., 2024; Hooper et al., 2024) and low-rank approximation (Chang et al., 2025), primarily reduce the per-token memory footprint. However, they do not address the cache's inherent linear growth with context length and quadratic complexity of attention, and thus remain insufficient for truly long-context inference. We therefore focus on KV eviction, which directly bounds memory by retaining only the most valuable tokens.

KV eviction faces two key challenges. First, it requires accurately predicting which tokens will remain important for future decoding. Most existing strategies are training free, relying on hand crafted proxy scores derived from simple static statistics or strong modeling assumptions, an approach that often fails to capture the richer, context dependent patterns governing future token usage. For exam-

---

[*]Equal contribution. [†]Corresponding authors. Contact emails: Xintong Yang <youngzncu1010@gmail.com>, Hao Gu <marcusguhao@gmail.com>. [1]The Hong Kong University of Science and Technology [2]Zhejiang University. Correspondence to: Sirui Han <siruihan@ust.hk>, Yike Guo <yikeguo@ust.hk>.

*Proceedings of the 43rd International Conference on Machine Learning*, Seoul, South Korea. PMLR 306, 2026. Copyright 2026 by the author(s).

ple, SnapKV (Li et al., 2024) estimates importance from a local attention window, which can bias retention toward nearby tokens; KeyDiff (Park et al., 2025) uses inter-key dissimilarity as a proxy for informativeness; and Expected Attention (Devoto et al., 2025) scores keys using "average" queries computed from corpus-level query statistics, an assumption that may not adapt to the specific context and query distribution encountered at inference time. Second, existing methods permanently discard evicted tokens, which cannot be retrieved if they become relevant later, incurring irreversible information loss. To address these challenges, we propose a learnable indexer that more accurately estimates token importance, coupled with a dedicated memory module that preserves information from evicted tokens for later retrieval.

To summarize, our contributions are as follows:

1. We propose **a learnable indexer** that more accurately predicts the importance of KV tokens and enables adaptive KV eviction.

2. To mitigate irreversible forgetting caused by permanently discarding evicted tokens, we introduce a **memory module** that compresses evicted tokens into a fixed-size, compact latent memory, which is **updated online** during inference.

3. We conduct comprehensive experiments demonstrating that our method consistently improves long context performance, achieving strong gains on RULER across Qwen/Mistral/Llama (up to +25 points), more robust NIAH retrieval, and better LongBench score.

## 2. Background

### 2.1. Attention and KV Caching

The Transformer architecture (Vaswani et al., 2017) serves as the foundational component of modern LLMs. It processes an input sequence $X = (x_1, x_2, \ldots, x_T) \in \mathbb{R}^{T \times d_{\text{model}}}$ through stacked Transformer blocks. Each block $f(\cdot)$ sequentially applies causal self-attention and a feed-forward network (FFN) to produce the output sequence $X' = (x_1', x_2', \ldots, x_T') \in \mathbb{R}^{T \times d_{\text{model}}}$:

$$X' = f(X) = \text{FFN}(\text{Attention}(X)). \quad (1)$$

**Causal self-attention.** Given hidden states $X$, self-attention projects each token into query, key, and value vectors using projection matrices $W_q, W_k, W_v$:

$$Q = XW_q, \quad K = XW_k, \quad V = XW_v. \quad (2)$$

The attention output is then computed as:

$$O^{\text{attn}} = \text{Softmax}\left(\frac{QK^\top}{\sqrt{d_{\text{model}}}} + Mask\right)V = AV, \quad (3)$$

where $O^{\text{attn}} \in \mathbb{R}^{T \times d_{\text{model}}}$ denotes the output matrix, $A \in \mathbb{R}^{T \times T}$ is the attention matrix, and $Mask$ is a causal mask with upper-triangular entries set to $-\infty$ to prevent future token access.

**KV Caching.** During autoregressive decoding, recomputing keys and values for all prior tokens $x_1, \ldots, x_T$ when processing a new token $x_{T+1}$ is computationally inefficient. KV caching circumvents this redundancy by maintaining a cache $\mathcal{C} = \{(k_j, v_j)\}_{j=1}^T$ of previously computed features. For the new token, only its query, key, and value vectors $(q_{T+1}, k_{T+1}, v_{T+1})$ are generated. The attention output for $x_{T+1}$ is derived by aggregating the values from the updated cache using attention weights computed against the query:

$$o_{T+1}^{\text{attn}} = \sum_{j=1}^{T+1} \text{softmax}\left(\frac{q_{T+1}K_{1:T+1}^\top}{\sqrt{d_{\text{model}}}}\right)_j v_j$$

where the Softmax is normalized over the causal context $j \in \{1, \ldots, T+1\}$. This formulation highlights that the output is a weighted sum of cached values. While KV caching substantially reduces decoding latency, the linear growth of $\mathcal{C}$ with context length makes it the dominant memory consumer during long-context inference.

### 2.2. Formulation of KV cache Eviction Methods

We can formulate most KV cache eviction methods in a unified scoring-and-selection framework. Given the cached keys and values $K, V \in \mathbb{R}^{B \times H_{kv} \times L \times d_{\text{head}}}$, an eviction policy first assigns an importance score to each cached token position:

$$S = f(\cdot) \in \mathbb{R}^{B \times H_{kv} \times L},$$

where $S_{b,h,t}$ measures the importance of token $t$ for head $h$ in sample $b$. The policy then keeps the top-$L'$ tokens and compacts the cache by gathering along the sequence dimension:

$$\mathcal{I} = \text{TopK}(S, L'), \ [K', V'] = \text{Gather}(K, V, \mathcal{I}).$$

Here $\mathcal{I}$ denotes the indices of retained tokens and $\text{Gather}(\cdot)$ selects entries along the token dimension.

Different methods mainly differ in the definition of the score function $f(\cdot)$. SnapKV uses attention from the most recent window of queries to score past keys. Concretely, it defines the per-key importance as the average attention mass assigned to key position $t$ by the last $w$ queries:

$$S_t = \frac{1}{w} \sum_{i=L-w}^{L-1} \text{softmax}\left(\frac{QK^\top}{\sqrt{d}}\right)_{i,t}$$

Knorm: scores tokens by key magnitude, typically

$$S_t = \|K_t\|_2,$$

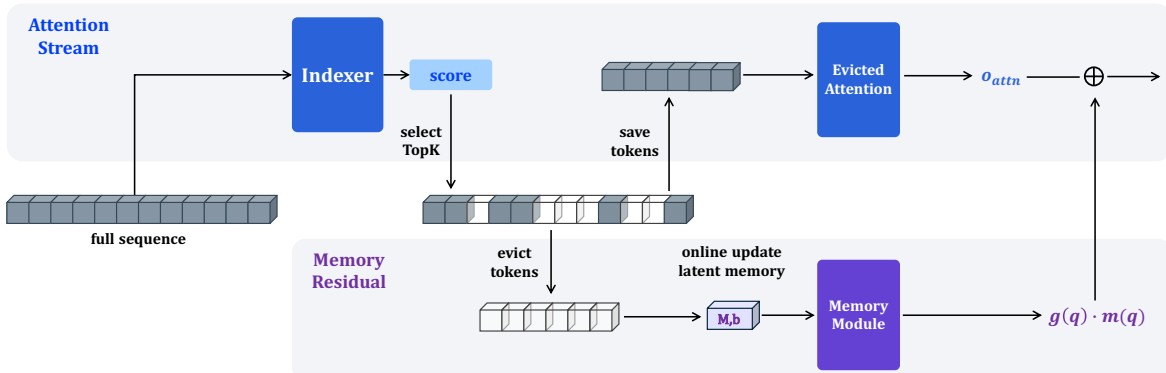

*Figure 1.* The overall pipeline of **IndexMem** is as follows: in the main attention stream, we use the learnable indexer to accurately select which KV tokens to save and evict the rest. The evicted tokens are then used to **update a latent memory online**, and the memory readout is added as **a residual to compensate** the main attention stream for the information lost due to eviction.

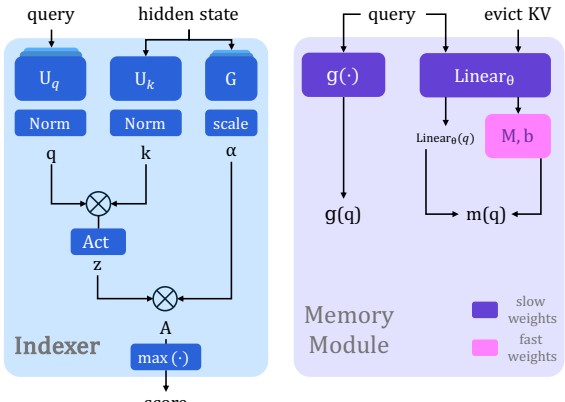

*Figure 2.* **Architectures of the Indexer and the Memory module.** **Left:** the Indexer adopts an MQA-style design with norm on both multi head **q** and single head cached **k** (**k** are continuously cached during decoding); it computes token scores via gated **q**, **k** similarity, followed by max aggregation. **Right:** the Memory module produces a residual readout $m(q)$ from a fixed-size latent state. Evicted tokens update the fast weights $(\mathbf{M}, b)$ online, while $g(\cdot)$ and $\text{Linear}_\theta(\cdot)$ are slow weights learned during training.

so tokens with smaller key norms are considered less important and are evicted first. We can plug in other eviction rules (e.g., TOVA (Oren et al., 2024), KeyDiff, Expected Attention) by specifying their corresponding score function $S = f(\cdot)$ under this same framework.

## 3. Methods

### 3.1. Learnable Indexer for Token Importance

The core challenge in KV eviction is accurately estimating token importance through a proxy. Existing methods are limited by heuristic designs and a tendency to overemphasize local tokens. Motivated by the lightning indexer in DeepSeek Sparse Attention (Liu et al., 2025), we introduce a lightweight, learnable **indexer** to assess token importance

for KV retention. Given hidden states $X \in \mathbb{R}^{L \times d_{\text{model}}}$ and (pre-RoPE) query states $Q \in \mathbb{R}^{H \times L \times d_{\text{head}}}$, the indexer outputs a dense query-to-key score matrix

$$A = \text{Indexer}(X, Q) \in \mathbb{R}^{L \times L},$$

where $A_{s,t}$ measures how important the key token at position $t$ is to the query at position $s$.

**Architecture.** We first construct QK-Norm (Henry et al., 2020) features for stable training. Let $H_{\text{index}}$ and $d_{\text{index}}$ denote the number of indexer heads and the per-head dimension, respectively, where typically $H_{\text{index}} \ll H$ and $d_{\text{index}} \ll d_{\text{head}}$ to reduce computation. We obtain indexer queries **q** by down-projecting the multi-head query at position $s$:

$$\mathbf{q}_s = U_q \, \text{flatten}(Q_s) \in \mathbb{R}^{H_{\text{index}} d_{\text{index}}},$$

and reshape $Q_s$ as per-head features $Q_{s,h} \in \mathbb{R}^{d_{\text{index}}}$. Keys **k** are derived from hidden states: $\mathbf{k}_t = U_k X_t \in \mathbb{R}^{d_{\text{index}}}$. Following an MQA-style (Shazeer, 2019) design, the indexer uses a single shared key **k** for all head. Then, we apply RMSNorm to obtain QK-Norm features: $\mathbf{q} = \text{Norm}(\mathbf{q})$, $\mathbf{k} = \text{Norm}(\mathbf{k})$.

We add a learnable gate to modulate the contribution of indexer heads and improve expressiveness. For each position $s$, we compute a head-gating vector

$$\alpha_s = \frac{G X_s}{\sqrt{H_{\text{index}} d_{\text{index}}}} \in \mathbb{R}^{H_{\text{index}}}.$$

Given a query position $s$ and a key position $t$, let $\mathbf{q}_s \in \mathbb{R}^{H_{\text{index}} \times d_{\text{index}}}$ be the query features and $\mathbf{k}_t \in \mathbb{R}^{d_{\text{index}}}$ the shared key feature. We compute the per-head similarities as **z** and aggregate them with the gate:

$$\mathbf{z}_{s,t} = \text{act}\,(\mathbf{q}_s, \mathbf{k}_t), \quad A_{s,t} = \alpha_s^\top \mathbf{z}_{s,t} + Mask_{s,t},$$

where $Mask_{s,t}$ is the causal mask. The resulting score matrix $A$ serves as a learnable proxy for token importance. We

reduce scores over query set $\mathcal{Q}$ to obtain per-token importance for KV retention:

$$\text{imp}_t = \max_{s \in \mathcal{Q}} A_{s,t}$$

We define the query set $\mathcal{Q}$ used for importance aggregation as follows: during prefill, the indexer uses all queries in the prompt; during decoding, it aggregates over the queries generated within each compression interval.

This indexer design offers several advantages. During decoding, it reuses the growing $\mathbf{k}$ cache to predict token importance. Unlike methods such as Expected Attention and SnapKV that typically materialize all KV pairs then evict, our indexer enables **pre-eviction**: before prefill, it uses score to predict which KV entries should be retained, so the subsequent prefill computes and caches only the selected KV states. The overall architecture is illustrated in Figure 2.

**Training.** Our proposed MQA-style indexer is a lightweight attention-like module that predicts the backbone attention scores. During training, we freeze the backbone LLM and optimize only the learnable indexer. Given a sequence of length $L$, the indexer outputs a score map $A \in \mathbb{R}^{L \times L}$, where $A(q, k)$ are logits over keys for each query $q$. We distill it to match the backbone's attention behavior by aligning pooled per-key importance distributions derived from the teacher and the indexer. Concretely, we use the teacher attention logits $T = QK^\top / \sqrt{d_{\text{model}}}$ and apply a max aggregation over queries: a key is deemed important if it is highly scored by any query. We then minimize

$$D_{\text{KL}}\Big(\text{softmax}\big(\max_q T(q, \cdot)\big) \,\Big\|\, \text{softmax}\big(\max_q A(q, \cdot)\big)\Big).$$

To stabilize optimization, we exclude **sink tokens** (Xiao et al., 2023) from the KL loss by masking out a fixed set of sink positions on the key axis, since their consistently large attention weights can dominate gradients.

Naively materializing the full teacher score matrix ($T$) is prohibitively memory-intensive, as it requires storing $O(L^2)$ entries. Instead, we compute the pooled vectors $\max_q T(q, \cdot)$ and $\max_q A(q, \cdot)$ in a streaming (FlashAttention-style) manner: we iterate over queries in chunks and stream over keys, updating an $O(L)$ running-max vector over keys, never instantiating the full $L \times L$ matrix. Finally, we apply a numerically-stable softmax on the key axis and compute the KL exactly for the pooled distributions while preventing materializing the full attention map.

### 3.2. Latent Memory for Evicted Tokens

Most KV cache compression methods rely on an **evict or keep** decision: once a token is evicted, its KV states are permanently discarded. This design is often sufficient for **retrieval style** benchmarks (e.g., needle-in-a-haystack/RULER), where preserving a small set of evidence

tokens is enough. However, for **holistic long-context reasoning** (e.g., QA and summarization), useful information can be spread across many seemingly low saliency tokens. Aggressive eviction therefore introduces an irreversible information loss that accumulates over time.

Existing approaches to mitigate forgetting in KV cache eviction have explored reconstruction based methods like KVReviver (Yuan et al., 2025a), which rebuild KV from compact sketches, yet suffer from error amplification during softmax. Offloading alternatives (e.g., NOSA (Huang et al., 2025), InfLLM (Xiao et al., 2024)) preserve tokens in CPU memory, yet they introduce throughput bottlenecks due to the high latency of CPU-GPU data transfers.

To address these problems, we compact evicted tokens into a fixed-size latent memory. Our memory design consists of two components: **how to write** evicted information into memory, and **how to read** it back to compensate attention.

**Why not memory-as-tokens.** A straightforward approach is to write evicted KV into a small set of latent KV tokens ("memory-as-tokens") and let standard softmax attention read both retained tokens and latent memory. In practice, token placement is tricky: putting latent tokens at the prefix often turns them into attention sinks, that attract large attention mass. While placing them later makes them unable to access for early queries. A simple fix is a special attention mask that makes latent tokens globally visible to all queries. More fundamentally, this formulation can be brittle: the latent tokens may collapse to a general "mean" summary, and softmax can amplify small mismatch.

**Readout as residual compensation.** Instead of injecting latent memory as additional KV tokens inside softmax, we treat memory as an explicit compensation residual for the information removed by eviction. Concretely, we augment the original attention output with a gated memory readout:

$$o = o_{\text{attn}} + g(q) \cdot m(q), \tag{4}$$

where $o_{\text{attn}}$ is computed only over the retained KV cache. The memory readout $m(q)$ summarizes useful signals from evicted tokens conditioned on the current query $q$. The gate $g(q) \in [0, 1]$ controls whether and how much the model should rely on the latent memory for current query, enabling a safe fallback that $g(q) = 0$.

**Slow and fast weights in the memory module.** We use one latent memory module per layer, shared across all attention heads. The module consists of (i) slow weights $\theta$, updated by gradients during training, and (ii) fast weights, updated online at inference time by simple update rules. Using slow weights alone often collapses to a dataset-specific mean compensation for evicted attention. The fast weights maintain a fixed-size state matrix $\mathbf{M} \in \mathbb{R}^{d_{\text{mem}} \times d_{\text{model}}}$ and

a stabilizer vector $b \in \mathbb{R}^{d_{\text{mem}}}$, where $d_{\text{mem}}$ is the memory-state dimension. Given a query $q \in \mathbb{R}^{d_{\text{model}}}$, we compute a projection $\text{Linear}_\theta(q) \in \mathbb{R}^{d_{\text{mem}}}$ and read from memory as

$$m(q) \ = \ \frac{\text{Linear}_\theta(q)^\top \mathbf{M}}{\text{Linear}_\theta(q)^{\odot 2 \, \top} b + \epsilon}, \tag{5}$$

where $\text{Linear}_\theta(q)^{\odot 2} = \text{Linear}_\theta(q) \odot \text{Linear}_\theta(q)$ and $\epsilon$ is a small constant.

**Online write as fast-weight updates.** The memory state $(\mathbf{M}, b)$ is updated online per evict step. Given the evicted set $\mathcal{E}$ with key-value pairs $\{(k_i, v_i)\}_{i \in \mathcal{E}}$, we apply an outer-product accumulation with decay:

$$\mathbf{M} \leftarrow \lambda \mathbf{M} \ + \ \eta \sum_{i \in \mathcal{E}} \text{Linear}_\theta(k_i) \, v_i^\top, \tag{6}$$

$$b \leftarrow \lambda b \ + \ \eta \sum_{i \in \mathcal{E}} \text{Linear}_\theta(k_i) \odot \text{Linear}_\theta(k_i), \tag{7}$$

where $\lambda \in (0, 1]$ is a decay factor and $\eta > 0$ is the write strength.

**Slow-weight training.** We train the slow-weight components (the projection $\text{Linear}_\theta(\cdot)$ and gate $g(\cdot)$) by gradient to make the memory readout match the missing residual introduced by eviction. Concretely, we minimize an MSE objective

$$\mathcal{L}_{\text{mem}} \ = \ \big\| o - o_{\text{attn}} - g(q) \cdot m(q) \big\|_2^2, \tag{8}$$

where $o$ is the full attention output and $o_{\text{attn}}$ is the output computed from the compressed KV cache.

# 4. Experiment

**Models and Evaluation Setup.** We conduct experiments on three representative LLM backbones: Qwen3-8B (Yang et al., 2025), Mistral-7B-v0.3 (Jiang et al., 2023), and Llama-3.1-8B-Instruct (Grattafiori et al., 2024). Following common practice, we define the compression ratio $r \in [0, 1]$ such that each method evicts an $r$ fraction of cached KV entries and retains the top $(1 - r)$ fraction according to its token-importance scores. To avoid degenerate behaviors caused by sink tokens, we additionally keep the first four tokens in the cache for all methods. We implement IndexMem and all baselines within the same evaluation pipeline using the KVPRESS (NVIDIA Corporation, 2025) GitHub repository, ensuring consistent eviction protocols and fair comparisons across methods. Unless otherwise specified, all experiments are run on a single NVIDIA H800 GPU.

**Inference schedule and cache budget.** Our evaluation primarily targets the long-prefill, short-decode regime. After finishing the prefill pass over a prompt of length $L$, we immediately compute token importance, perform one-shot compression, and write evicted KV pairs into the latent memory. Concretely, we compress the prefill KV cache to a fixed fraction $(1 - r)L$ (i.e., evict an $r$ portion of tokens). During decoding, we further apply periodic compression every $\tau = 128$ generated tokens to prevent the cache from growing. At each compression, we score tokens using all queries observed since the previous compression and retain the most important entries under a fixed KV budget $B_{\max}$ (implementation-wise, $B_{\max}$ can be set to $(1 - r)L$ plus the most recent local window).

**Training Setup**

**Indexer and memory hyperparameters.** The indexer uses $H_{\text{index}} = H/4$ heads, where $H$ is the number of attention heads in the backbone LLM. Its down-projection dimension is set to $d_{\text{index}} = d_{\text{head}}/8$ (with $d_{\text{head}}$ the per-head hidden dimension), and $G$ is implemented as a lightweight per-head gating module. For the memory module, we set the latent dimension to $d_{\text{mem}} = d/8$, where $d_{\text{model}} = H \cdot d_{\text{head}}$ denotes the model hidden size.

**Training protocol.** We freeze the backbone LLM and adopt a two-stage training scheme. (i) We first train the indexer alone, so that it learns to estimate token importance and make reliable evict decisions. (ii) We then jointly train the indexer and the memory module, where the latent memory is optimized using the evicted-attention output as the learning signal, encouraging it to compensate for information lost due to eviction. We conduct SFT on LongAlpaca (Chen et al., 2024) with a chunk-wise KL objective, and use DDP (without context parallelism) for training. We use a warmup-stable-decay (WSD) learning-rate schedule with 100 warmup steps to $1 \times 10^{-3}$, followed by 2000 stable steps and 2000 decay steps to $7.5 \times 10^{-6}$.

**Baselines.** We compare our method (IndexMem) with representative KV-cache eviction baselines, all implemented and evaluated under the same framework for a fair comparison. Specifically, we include ExpectedAttention (attention-statistics-based importance), KeyDiff (key-difference saliency), TOVA (Token Omission Via Attention; retrieval-oriented eviction), and two widely used heuristic compressors, SnapKV and PyramidKV.

**RULER results.** We evaluate long-context capability on the RULER suite (Hsieh et al., 2024) under both RULER-4K and RULER-16K settings. In our implementation, each subtask is scored by `string_match`. We report the overall RULER score as an unweighted average over all subtasks in the evaluation set. Table 1 shows that IndexMem is consistently the most robust method under KV eviction. Under mild compression ($r \leq 0.25$), IndexMem nearly matches the full-cache baseline across all backbones, in-

| Model | Method | Ruler 4k | | | | | | Ruler 16k | | | | | |
|-------|--------|-----|-----|-----|-----|-----|-----|-----|-----|-----|-----|-----|-----|
| | | 0% | 10% | 25% | 50% | 75% | 90% | 0% | 10% | 25% | 50% | 75% | 90% |
| *Qwen3-8B* | IndexMem (ours) | 95.3 | **93.8** | **90.3** | **78.9** | **72.9** | **46.2** | 92.9 | **91.6** | **86.8** | **80.0** | **72.2** | **56.0** |
| | KeyDiff | 95.3 | 93.8 | 89.4 | 78.6 | 64.4 | 37.9 | 92.9 | 88.9 | 82.9 | 74.5 | 66.9 | 53.1 |
| | TOVA | 95.3 | 89.0 | 82.5 | 77.6 | 62.4 | 24.7 | 92.9 | 88.3 | 81.7 | 76.2 | 68.7 | 52.4 |
| | SnapKV | 95.3 | 92.6 | 84.0 | 55.7 | 33.1 | 19.2 | 92.9 | 90.1 | 81.5 | 62.8 | 41.7 | 26.8 |
| | PyramidKV | 95.3 | 91.4 | 75.1 | 40.8 | 29.7 | 19.5 | 92.9 | 89.2 | 76.1 | 47.8 | 32.6 | 22.0 |
| *Mistral-7B* | IndexMem (ours) | 92.1 | **88.6** | **85.0** | **82.8** | **68.8** | **41.5** | 86.1 | **81.0** | **75.9** | 68.5 | **66.0** | **52.9** |
| | ExpectedAttention | 92.1 | 77.1 | 66.3 | 44.8 | 28.1 | 13.2 | 86.1 | 69.5 | 59.3 | 41.0 | 26.2 | 15.4 |
| | TOVA | 92.1 | 83.1 | 78.4 | 75.5 | 62.0 | 29.2 | 86.1 | 77.7 | 73.8 | **70.1** | 61.0 | 47.7 |
| | SnapKV | 92.1 | 71.5 | 57.6 | 41.3 | 31.9 | 19.8 | 86.1 | 62.3 | 49.5 | 36.5 | 27.1 | 21.3 |
| | PyramidKV | 92.1 | 70.8 | 57.1 | 38.6 | 29.6 | 19.9 | 86.1 | 53.4 | 45.0 | 33.2 | 22.9 | 18.0 |
| *Llama3.1-8B* | IndexMem (ours) | 95.3 | 95.6 | **95.6** | **93.3** | 78.3 | **55.2** | 93.4 | **93.5** | **92.9** | **88.7** | **74.3** | 56.0 |
| | ExpectedAttention | 95.3 | **95.7** | 95.3 | 92.2 | 75.9 | 30.6 | 93.4 | 93.4 | 92.8 | 86.0 | 66.4 | 25.5 |
| | TOVA | 95.3 | 93.2 | 87.3 | 76.2 | 63.3 | 37.5 | 93.4 | 90.9 | 86.1 | 77.9 | 68.4 | **59.2** |
| | SnapKV | 95.3 | 95.5 | 88.8 | 81.8 | 63.2 | 43.4 | 93.4 | 89.4 | 82.0 | 68.0 | 43.1 | 25.6 |
| | PyramidKV | 95.3 | 84.5 | 79.4 | 65.8 | 43.6 | 23.1 | 93.4 | 85.2 | 77.5 | 62.5 | 39.6 | 24.1 |

*Table 1.* Ruler results under different compression ratios.

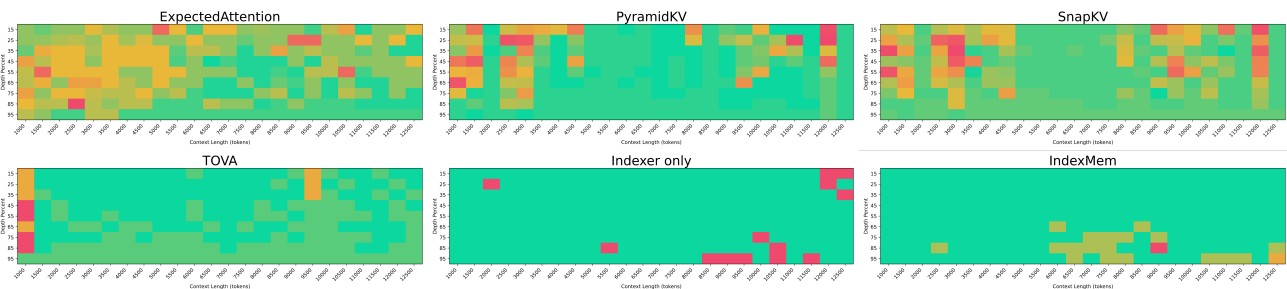

*Figure 3.* Needle-in-a-Haystack (NIAH) heatmaps of Llama-3.1-8B-Instruct under KV eviction at 50% compression ratio.

dicating that the learned indexer can precisely remove a substantial fraction of unnecessary tokens with minimal accuracy loss. As eviction becomes aggressive ($r \geq 0.5$), the gap widens: heuristic methods (e.g., SnapKV/PyramidKV) degrade rapidly, while IndexMem degrades more gracefully and preserves substantially higher scores, especially at extreme budgets ($r = 0.75$–$0.9$). This effect is most pronounced in longer contexts (RULER-16K), demonstrating the effectiveness of our learning-based indexer.

**NIAH results.** To visualize retrieval robustness across context lengths and needle positions, we additionally run the Needle-in-a-Haystack (NIAH) stress test (Kamradt, 2023). Figure 3 visualizes retrieval robustness under aggressive KV eviction. Overall, IndexMem exhibits the most stable behavior across the full sweep of context lengths (1K–12.5K) and needle depths (15%–95%), indicating that it can reliably recover the needle regardless of where it appears. In contrast, heuristic compression baselines (e.g., PyramidKV and SnapKV) show clear position-dependent failure modes: they perform reasonably when the needle lies in more favorable regions, but degrade sharply for harder placements, consistent with a strong recency bias and coarse token-selection granularity. Finally, comparing Indexer only vs. IndexMem, the learned indexer already achieves high retrieval quality

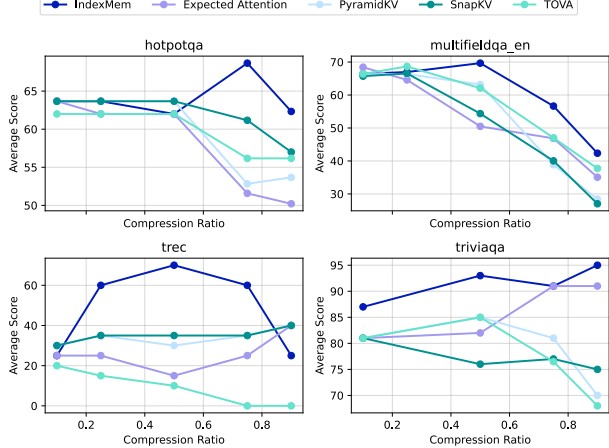

*Figure 4.* Scores on LongBench for Llama-3.1-8B-Instruct.

but exhibits rare catastrophic misses (isolated red cells). Adding the latent memory substantially reduces these failures, supporting our hypothesis that the memory residual compensates for information irreversibly lost due to eviction and improves worst-case retrieval robustness.

**LongBench results.** We further evaluate long-context understanding on LongBench (Bai et al., 2024) under KV-

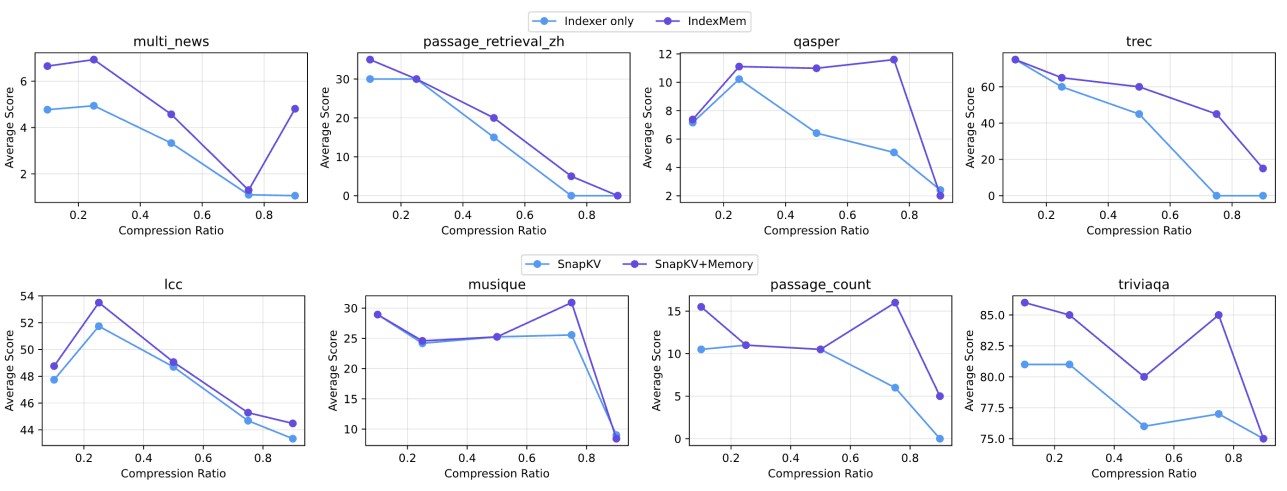

*Figure 5.* Ablation of Memory module on Llama-3.1-8B-Instruct of Longbench.

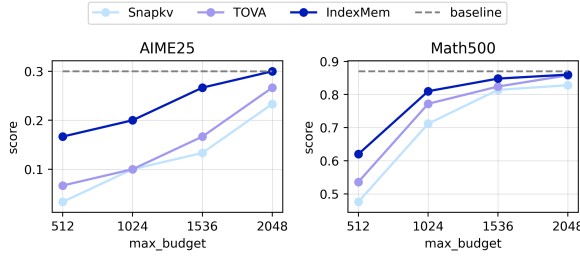

*Figure 6.* Decoding time KV cache compression on Qwen3-8B.

cache eviction. Figure 4 reports score–compression curves on representative longbench tasks. Overall, IndexMem is the most robust across LongBench tasks, degrading more gracefully as compression increases. It consistently leads on hotpotqa and multifieldqa_en, while PyramidKV/SnapKV drop sharply at $r \geq 0.75$. On triviaqa, IndexMem remains strong even at extreme budgets and can sometimes improve with higher compression. We attribute this to an information-density effect because moderate eviction removes low-value or distracting tokens and sharpens the retained context. This effect is not isolated to triviaqa—the all-task LongBench average in Appendix C.3 also peaks at 50% compression for both IndexMem and PyramidKV before degrading under aggressive eviction. trec is more sensitive at $r = 0.90$, where performance may drop due to pruning task-critical conditioning tokens; notably, TOVA collapses under high compression. Overall, IndexMem offers a better accuracy–memory trade-off than prior eviction heuristics.

**Decoding Compression.** During decoding, we compress every $\tau = 128$ generated tokens. At each compression step, we enforce a maximum KV budget $B_{\max}$. Figure 6 reports results on AIME25 and Math500 by sweeping $B_{\max} \in \{512, 1024, 1536, 2048\}$. As expected, larger budgets consistently improve performance for all methods and

gradually approach the full-cache baseline. Importantly, IndexMem achieves the best accuracy under the same $B_{\max}$ across both benchmarks, demonstrating that learned importance estimation is especially beneficial when the KV cache is heavily constrained. Under moderate budgets ($B_{\max} \geq 1536$), the gap narrows as all methods retain enough context to recover most performance, but IndexMem remains consistently on top, indicating a better accuracy–memory trade-off throughout.

**Ablation.** We conduct ablations to isolate the effect of the latent memory module. First, we compare Indexer only (without memory compensate) against IndexMem (Indexer + memory residual). Across LongBench tasks, adding memory yields consistent gains, with the largest improvements under aggressive compression (e.g., $r \geq 0.5$). In particular, the memory residual substantially reduces catastrophic failures where Indexer only collapses at high compression (e.g., on trec and qasper QA tasks), indicating that storing evicted tokens and reading them back is crucial for mitigating irreversible forgetting introduced by eviction. Second, we evaluate whether the memory module is tied to our indexer or can generalize to other eviction policies. To this end, we plug the same memory module into SnapKV, forming SnapKV+Memory, where evicted tokens selected by SnapKV are used to update the memory online and the memory readout is added as a residual. We observe consistent improvements over SnapKV on multiple tasks, showing that the proposed latent memory is largely orthogonal to the choice of eviction heuristic and can serve as a drop-in component to recover information lost by KV eviction.

**Parameter and Efficiency Analysis.** We introduce only a small number of additional parameters. On Llama-3.1-8B-Instruct, the indexer adds 19.92M parameters, while the latent memory module is lightweight, contributing only

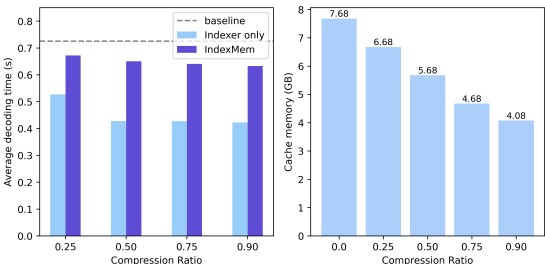

*Figure 7.* Efficiency analysis of Llama-3.1-8B-Instruct.

0.52M parameters. We measure efficiency under a long-context setting with 32K prefill and 1K decoding. Figure 7 reports (i) average decoding time and (ii) cache memory usage as a function of the compression ratio. As shown in the left panel, the indexer-only variant is consistently faster than the full-cache baseline, and decoding time decreases as eviction becomes more aggressive. Adding the memory module incurs a test-time update overhead, but still remains competitive and close to the baseline in latency across all compression ratios. The right panel shows that cache memory decreases monotonically with higher compression, dropping from 7.68 GB at $r = 0$ to 4.08 GB at $r = 0.9$. Here, cache memory accounts for the KV cache, the indexer key cache, and the latent memory state.

## 5. Related Work

**Sparse Attention.** A large body of work improves long-context efficiency by sparsifying attention computation, either by imposing structured patterns or by dynamically restricting which key-value (KV) blocks are accessed. Recent system- and kernel-oriented approaches aim to reduce memory movement and attention FLOPs without necessarily changing the model weights. Quest introduces query-aware KV page selection, estimating page criticality from lightweight metadata and loading only top-ranked pages during attention computation (Tang et al., 2024). MInference accelerates the prefilling stage via dynamic sparse attention, exploiting recurring sparse patterns (e.g., A-shape / vertical-slash / block-sparse) with head-wise offline pattern assignment and efficient GPU kernels (Jiang et al., 2024). MoBA applies a mixture-of-experts style routing to block attention, enabling a smooth trade-off between full and sparse attention for long contexts (Lu et al., 2025). NSA designs natively trainable sparse attention mechanisms aligned with modern hardware, emphasizing both training viability and inference efficiency (Yuan et al., 2025b).

**KV Cache Compression.** KV cache compression methods broadly fall into two categories that are often orthogonal and composable. Representation compression reduces the per-token footprint of KV states, e.g., low-bit quantization schemes tailored to the distributional properties of key-/

value tensors (KIVI (Liu et al., 2024), ZipCache (He et al., 2024)). In contrast, token reduction directly bounds KV memory by retaining only a subset of tokens via eviction or selection policies. Most existing eviction strategies are training-free and rely on heuristics or simple statistics. H2O retains "heavy hitter" tokens with large accumulated attention contributions, balancing them with recency to stabilize decoding (Zhang et al., 2023). SnapKV leverages a short observation window near the end of the prompt to infer per-head salient KV positions, which can introduce locality bias when long-range dependencies dominate (Li et al., 2024). To avoid explicit attention-score materialization, KeyDiff selects tokens based on key similarity/diversity as a proxy for importance (Park et al., 2025), while Expected Attention estimates KV importance by modeling how future query distributions would attend to past keys (Devoto et al., 2025). More recently, a line of work seeks better alignment with true decoding-time queries, e.g., generating pseudo lookahead queries to improve eviction consistency (Wang et al., 2025).

**Memory.** A common approach to extend an LLM's effective context is to introduce external memory via agentic retrieval (Hu et al., 2025) or RAG-style (Qian et al., 2025) pipelines, which store past information in a separate datastore and retrieve relevant chunks at inference time. While effective, these methods typically rely on an additional retriever, incur extra system complexity, and can be sensitive to retrieval errors or latency. Another line of work integrates memory inside the model through test-time adaptation or fast-weight mechanisms. For example, Titans (Behrouz et al., 2024) maintain an online-updated memory state to accumulate information beyond the attention window, and Product Key Memory (PKM) (Lample et al., 2019) and its fast-weight variants (e.g., FwPKM) (Zhao & Jones, 2026) provide large-capacity associative memories with learned addressing. More recently, test-time training methods such as TTT-E2E (Tandon et al., 2025) update a compact set of parameters or internal states online to absorb long-range information during generation.

## 6. Conclusion & Limitation

We present IndexMem, a learnable KV-cache eviction framework for long-context LLM inference. Our method introduces a learnable indexer to more accurately predict KV-token importance. To mitigate the irreversible forgetting caused by discarding evicted tokens, we further propose an online-updated latent memory module whose residual readout compensates the main attention stream. Together, these components demonstrate the promise of learnable architectures for efficient attention and offer a favorable accuracy–memory trade-off across long-context benchmarks.

## Impact Statement

This paper presents work whose goal is to improve the efficiency of long-context large language model inference. By reducing the memory footprint of the KV cache, IndexMem may help lower the hardware cost of serving long-context models and make such systems more accessible under limited computational resources. This can also potentially reduce the energy and infrastructure requirements associated with long-context inference.

At the same time, improving inference efficiency may also make large language models easier to deploy at scale, including in settings where model outputs could be unreliable, biased, or misused. In addition, KV-cache eviction and memory compression may affect which parts of the input are preserved, so care should be taken when applying such methods in high-stakes domains where missing a small but important detail could lead to harmful decisions. Our method is intended as a general efficiency technique rather than a solution to these broader safety and reliability challenges. We encourage practitioners to evaluate compressed long-context systems carefully under their target use cases before deployment.

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

# A. Limitations and Future Work.

Long chain-of-thought (CoT) reasoning has become a standard capability of modern language models. However, longer CoT also leads to substantially larger KV-cache memory consumption, which can become the dominant memory bottleneck during long-context inference. As a result, the focus of memory-efficient LLM research is gradually expanding from reducing model weights through parameter compression (Gu et al., 2025b; Li et al., 2025; 2026) and quantization (Xu et al., 2026; Gu et al., 2025a; 2026) toward reducing the KV-cache footprint. Although recent works have explored reducing reasoning length or adaptively controlling CoT generation (Chen et al., 2026; Wang et al., 2026; Zhu et al., 2026), our work focuses on a complementary direction: improving KV-cache efficiency while preserving the model's ability to reason over long contexts.

While effective, our design space remains far from fully explored. First, the training objectives and supervision signals used to learn token importance and memory updates may not be optimal. Alternative objectives could further improve robustness, particularly under extreme compression ratios where retaining a small number of critical tokens becomes especially challenging. Second, our training is conducted under limited token budgets and relatively modest-scale settings, and we have not yet fully evaluated large-scale training regimes or broader model families. Finally, our current method is trained as an add-on module while keeping the backbone model frozen. An important future direction is to endow backbone models with native eviction-and-memory capabilities through continual training, allowing efficient attention, eviction decisions, and memory writing/reading to be learned jointly in an end-to-end manner.

# B. Cross-layer Score Redundancy and Index Reuse

### B.1. Motivation: Cross-layer Redundancy

Layer-level token scores can be noisy, and token scoring quality can vary substantially across layers, meaning that some layers produce score distributions that are weakly discriminative, close to uniform, and therefore make top-$K$ selection unstable and sensitive to small perturbations. This motivates aggregating multiple layer-wise score signals to reduce variance.

A complementary observation is that important token sets are often partially shared across nearby layers. Although different layers may assign different score magnitudes, the resulting top-$K$ retained indices tend to have non-trivial overlap, especially among neighboring layers. This suggests that repeatedly computing an entirely independent token-importance signal for every layer may be unnecessary. Instead, one can exploit cross-layer redundancy either by *soft score aggregation*, which combines score vectors across layers, or by *hard index reuse*, which reuses selected indices or indexer scores within a short group of layers. In this appendix, we first use running mean as a diagnostic score-aggregation study, and then connect this observation to an IndexCache-style (Bai et al., 2026) score reuse strategy.

### B.2. Running Mean as Cross-layer Score Aggregation

**Baseline: single-layer token scoring.** Given a chosen layer, together with an implicit head aggregation rule, the compressor keeps the $K$ tokens with the largest scores in the layer-level vector $\mathbf{s}_\ell \in \mathbb{R}^T$, with ties broken deterministically. We refer to this as the *single-source scoring* baseline.

**Running Mean aggregation.** Let $\mathbf{s}_\ell \in \mathbb{R}^T$ denote a layer-level score vector, aggregated across heads within a layer by a fixed rule. The *running mean* computes an averaged score signal across layers:

$$\bar{\mathbf{s}}_m^{(\text{naive})} = \frac{1}{m} \sum_{\ell=1}^m \mathbf{s}_\ell, \qquad m = 1, \ldots, N_{\text{layer}},$$

where $N_{\text{layer}}$ denotes the number of transformer layers in the backbone. In practice, we use $\bar{\mathbf{s}}_{N_{\text{layer}}}^{(\text{naive})}$ to perform top-$K$ selection. Intuitively, averaging reduces variance when some layers provide noisy estimates, and it is especially meaningful when important token sets are partially shared across nearby layers.

**Why naive averaging can fail on spiky-dependency tasks.** Averaging is not always beneficial. When a task requires preserving a small set of highly critical tokens, a few layers may produce sharply peaked and accurate score distributions, while others remain diffuse. Naively averaging *mixes* diffuse, low-quality signals into the peaked signal, reducing the relative margin between truly important tokens and the rest. This degradation is most visible when some layers have not yet formed a confident preference over tokens.

**Key diagnostic: entropy of the score-induced distribution.** We use entropy as a proxy for how *confident* or *selective* a layer/head is. High entropy indicates a near-uniform distribution, i.e., weak discrimination, whereas low entropy indicates a more concentrated distribution, i.e., strong preference over a subset of tokens. Given a probability distribution $\mathbf{p} \in \Delta^{T-1}$, we use normalized entropy

$$\mathsf{H}(\mathbf{p}) = -\frac{1}{\log T} \sum_{t=1}^{T} p_t \log(p_t + \epsilon),$$

so that $\mathsf{H}(\mathbf{p}) \in [0, 1]$.

**From scores to a probability distribution.** Let $\mathbf{s} \in \mathbb{R}^T$ denote the token scores for a fixed layer/head. To compute entropy, we map $\mathbf{s}$ to $\mathbf{p} \in \Delta^{T-1}$.

**Softmax;**

$$p_t = \frac{\exp(s_t/\tau)}{\sum_{j=1}^{T} \exp(s_j/\tau)} \quad (t = 1, \ldots, T),$$

`softmax` with temperature $\tau > 0$. We implement the standard stabilization $\exp(s_t/\tau - \max_j s_j/\tau)$, which leaves $\mathbf{p}$ unchanged while improving numerical stability.[1]

**L1 normalization with conditional min-shift; `negonly`.** When using $p_t = \tilde{s}_t / \sum_j \tilde{s}_j$, we must ensure non-negativity. We apply a *conditional* min-shift:

$$\tilde{s}_t = \begin{cases} s_t - \min_j s_j, & \text{if } \min_j s_j < 0, \\ s_t, & \text{otherwise,} \end{cases} \qquad p_t = \frac{\tilde{s}_t}{\sum_{j=1}^{T} \tilde{s}_j + \epsilon}.$$

This "only-if-negative" shift preserves the score geometry when $\mathbf{s}$ is already non-negative, whereas an unconditional shift would alter $\mathbf{p}$ even when not required.

**Entropy-gated Running Mean: skipping high-entropy sources.** We now describe the main refinement. High entropy typically means the scorer has not formed a reliable preference over tokens yet, i.e., the score distribution is too flat to be predictive. Averaging such high-entropy scores into the running mean can *contaminate* the aggregated signal, making it less discriminative. Hence, we *skip* layers/heads whose entropy is above a threshold.

Let $\mathsf{H}_\ell$ be the normalized entropy computed from layer $\ell$, or from a chosen head statistic. Given a threshold $\gamma \in [0, 1]$, define an inclusion indicator

$$\alpha_\ell = \mathbb{I}\left[\mathsf{H}_\ell \leq \gamma\right].$$

Then the entropy-gated mean is

$$\bar{\mathbf{s}}^{(\text{skip-high})} = \frac{\sum_{\ell=1}^{N_{\text{layer}}} \alpha_\ell \mathbf{s}_\ell}{\sum_{\ell=1}^{N_{\text{layer}}} \alpha_\ell + \delta},$$

where $\delta > 0$ avoids division by zero when all layers are skipped. This is a variance-reduction strategy with a quality filter: we average only among layers that appear selective, i.e., low entropy, and avoid layers whose scores are likely under-confident.

**Additional variants.** We additionally considered: (i) **skip-high vs. skip-low**, which filters under-confident vs. highly selective sources, and (ii) **entropy computed with `softmax` vs. `negonly`**, which changes how scores are normalized into probabilities. Their results are included in Table 2.

**Experimental summary.** We evaluate Expected Attention (EA) and running-mean variants on *RULER*. We report the *overall average* score, computed as the mean over tasks, under compression ratios $\mathrm{CR} \in \{0.25, 0.50, 0.75, 0.90\}$. Table 2 shows that entropy-gated running mean with `skip-high`, computed via `softmax`, consistently improves over the naive layer-mean running mean, and is often competitive with or better than EA across moderate compression ratios. Interestingly, the best-performing variant can change at very high compression, $\mathrm{CR} = 0.90$, suggesting a different failure mode when the retained context becomes extremely sparse. Overall, these results support the view that cross-layer score signals contain reusable information.

---

[1]Softmax stabilization by subtracting the maximum is a standard numerical technique.

*Table 2.* Overall average score, computed as the mean over tasks, for Expected Attention (EA) and running-mean variants under different compression ratios (CR). Higher is better.

| Method | CR=0.25 | CR=0.50 | CR=0.75 | CR=0.90 |
|---|---|---|---|---|
| EA (`expected_attention`) | 89.19 | 76.53 | 54.53 | 27.78 |
| Layer-mean RM (`layer_mean`) | 81.43 | 74.00 | 51.98 | 31.93 |
| RM + entropy skip-high (`ent_skip_high`) | **89.71** | **79.08** | **57.76** | 32.93 |
| RM + entropy skip-low (`ent_skip_low`) | 82.32 | 73.38 | 54.07 | **33.13** |
| RM + peak gate (`peak_gate`) | 82.44 | 74.94 | 54.66 | 32.36 |
| RM + peak gate, keep-high (`peak_gate_keephigh`) | 83.27 | 74.84 | 52.79 | 31.02 |

*Table 3.* RULER results on Mistral-7B with IndexCache-style score reuse. Columns indicate context length and compression ratio. Higher is better. IndexMem here uses the updated training recipe of Section 4 together with IndexCache-style score reuse.

| Method | 4K-25% | 4K-50% | 4K-90% | 16K-25% | 16K-50% | 16K-90% |
|---|---|---|---|---|---|---|
| IndexMem + IndexCache (ours) | **92.4** | **90.0** | **76.1** | **82.5** | **79.3** | **66.8** |
| AdaKV | 72.8 | 56.7 | 22.8 | 68.6 | 52.1 | 23.8 |
| SnapKV | 57.6 | 41.3 | 19.8 | 49.5 | 36.5 | 21.3 |
| TOVA | 78.4 | 75.5 | 29.2 | 73.8 | 70.1 | 47.7 |

### B.3. From Running Mean to IndexCache

Running mean exploits cross-layer redundancy by *averaging score values*. However, when the top-$K$ token sets are already similar across neighboring layers, a simpler implementation is to reuse the score or index decision itself. This leads to an IndexCache-style strategy: instead of recomputing indexer scores independently for every layer, we compute the indexer scores once within a short group of neighboring layers and reuse the resulting scores or selected indices for the remaining layers in the group. In our implementation, we reuse indexer scores across every four consecutive layers.

This reuse strategy does not change the core token-selection objective of IndexMem. It only amortizes the overhead of score computation by exploiting the empirical redundancy of important token sets across nearby layers. Thus, running mean can be viewed as a soft aggregation diagnostic, while IndexCache-style reuse is a lightweight practical realization of the same cross-layer redundancy.

Table 3 reports RULER results on Mistral-7B with the IndexCache-style variant. These numbers are obtained with the updated training recipe described in Section 4 (WSD learning-rate schedule) combined with IndexCache-style score reuse, and therefore should not be directly compared to the Mistral-7B IndexMem entries in Table 1, which use a constant learning-rate schedule without IndexCache-style score reuse. The variant maintains strong accuracy under both 4K and 16K RULER settings, and compares favorably against AdaKV (Feng et al., 2026), SnapKV, and TOVA under matched compression ratios.

### B.4. Takeaway and Scope

The running-mean study is intended as an analysis of cross-layer score aggregation, rather than as the final IndexMem inference algorithm. Its main role is to show that layer-wise token-importance signals can contain reusable information: aggregating selective layers can stabilize token selection, while high-entropy layers may introduce noise. IndexCache-style reuse exploits the same property in a simpler and more practical way by reusing indexer scores or selected indices across nearby layers.

This observation also has limitations. Entropy is only a proxy for score reliability: low entropy does not guarantee correctness, and high entropy does not necessarily imply uselessness. The threshold $\gamma$ can be task- and model-dependent, and skipping too many layers may reduce the benefit of aggregation. Therefore, we treat running mean as a diagnostic aggregation study, while using IndexCache-style reuse as a lightweight practical variant for reducing indexer overhead without changing the core token-selection objective.

*Table 4.* Taxonomy of baselines considered in the main paper and this appendix.

| Category | Examples | Compression axis | Role in this paper |
|---|---|---|---|
| Learnable token eviction / retention | Locret, AdaKV | Sequence / token | Additional learned baselines |
| Representation-level KV compression | xKV | Per-token KV representation | Complementary comparison |
| Heuristic token eviction | SnapKV, PyramidKV, TOVA, EA | Sequence / token | Main reference baselines |
| Ours | IndexMem | Token eviction + latent memory | Proposed method |

*Table 5.* Per-task LongBench scores at 75% compression for IndexMem, xKV, and Locret. *Avg. (shown)* is the mean over the seven reported tasks and is *not* comparable to the all-task average in Table 6.

| Method | wikiqa | hotpotqa | triviaqa | passage_retrieval_en | multifieldqa_en | multi_news | multifieldqa_zh | Avg. (shown) |
|---|---|---|---|---|---|---|---|---|
| IndexMem (ours) | **50.97** | **68.67** | **91.00** | 40.00 | **59.80** | **25.40** | **56.58** | **56.06** |
| xKV | 46.21 | 55.81 | 90.00 | 1.00 | 32.61 | 25.04 | 46.13 | 42.40 |
| Locret | 10.10 | 13.20 | 59.55 | **85.07** | 16.94 | 19.13 | 16.51 | 31.50 |

# C. More Results

We provide additional experimental results that complement the main paper. These include a taxonomy of the additional baselines we compare against, per-task LongBench results at high compression with learnable and representation-level baselines, and aggregate LongBench results across compression ratios.

## C.1. Additional Baselines and Comparison Axes

The baselines used in the main paper and in this appendix fall into three groups along different compression axes. Table 4 summarizes this taxonomy and clarifies the role each baseline plays in our evaluation. In particular, low-rank / representation-level methods such as xKV operate on a different compression axis from token eviction and are therefore complementary, rather than directly competing, with our approach.

## C.2. Per-task LongBench Results at 75% Compression

We further compare IndexMem against a representation-level baseline (xKV) and a learnable retention baseline (Locret (Huang et al., 2024)) on seven representative LongBench tasks at 75% compression. Table 5 reports per-task scores; the last column reports the average over the seven *reported* tasks and should not be confused with the all-task LongBench average in Appendix C.3.

We highlight four observations. First, xKV reduces the *per-token* representation cost and is methodologically complementary to token selection; the comparison shows that IndexMem can match or exceed it on most information-dense tasks while operating on a different compression axis. Second, Locret is a learnable retention baseline that targets a similar problem to ours but tends to be brittle on multi-evidence reasoning tasks at high compression. Third, IndexMem is consistently stronger on information-dense QA and multi-evidence tasks, including *wikiqa*, *hotpotqa*, *triviaqa*, *multifieldqa_en*, and *multifieldqa_zh*. Finally, Locret is noticeably stronger on *passage_retrieval_en*, suggesting that retrieval-style tasks with a single localized answer span may favor different retention mechanisms; we therefore do not interpret these numbers as uniform dominance across all task types.

## C.3. Average LongBench Results across Compression Ratios

The main paper reports score–compression curves on a representative subset of LongBench tasks (Figure 4). This aggregate view shows that the advantage of IndexMem is not limited to the representative tasks visualized in the main paper.

IndexMem achieves the strongest average score at every reported compression ratio. Combined with the task-level curves in the main paper, this confirms that the advantage of IndexMem is not confined to a small set of selected LongBench tasks.

*Table 6.* Average LongBench score (over all tasks) across compression ratios. Higher is better.

| Method | 10% | 25% | 50% | 75% | 90% |
|---|---|---|---|---|---|
| IndexMem (ours) | **53.48** | **53.18** | **55.26** | **42.18** | **36.51** |
| Expected Attention | 43.78 | 43.88 | 42.09 | 36.33 | 28.60 |
| PyramidKV | 43.63 | 41.57 | 42.35 | 39.17 | 29.68 |
| SnapKV | 43.55 | 43.37 | 41.44 | 39.22 | 29.40 |
| TOVA | 43.81 | 43.08 | 42.10 | 39.05 | 32.56 |

## D. Notation

We summarize the key dimensions and tensor shapes used throughout the paper. Let $T$ denote the sequence length (or $L$ when focusing on the cached context), and $B$ denote the batch size. The Transformer hidden size is $d_{\text{model}}$, and the number of attention heads is $H$. The per-head dimension is

$$d_{\text{head}} = \frac{d_{\text{model}}}{H}. \tag{9}$$

For grouped-query / multi-query attention, we denote the number of key/value heads by $H_{\text{kv}}$ (with $H_{\text{kv}} \leq H$); thus cached keys/values have shape

$$K, V \in \mathbb{R}^{B \times H_{\text{kv}} \times L \times d_{\text{head}}}. \tag{10}$$

In the main text, token hidden states are denoted by $X \in \mathbb{R}^{L \times d_{\text{model}}}$, while (pre-RoPE) query states used by the indexer are denoted by

$$Q \in \mathbb{R}^{H \times L \times d_{\text{head}}}. \tag{11}$$

The indexer employs $H_{\text{index}}$ lightweight heads with per-head feature dimension $d_{\text{index}}$ (typically $H_{\text{index}} \ll H$ and $d_{\text{index}} \ll d_{\text{head}}$), and outputs a dense query-to-key score matrix

$$A = \text{Indexer}(X, Q) \in \mathbb{R}^{L \times L}. \tag{12}$$

Our latent memory module operates in the model space and maintains a fixed-size fast-weight state

$$M \in \mathbb{R}^{d_{\text{mem}} \times d_{\text{model}}}, \qquad b \in \mathbb{R}^{d_{\text{mem}}}, \tag{13}$$

where $d_{\text{mem}}$ is the memory-state dimension. Given a query vector $q \in \mathbb{R}^{d_{\text{model}}}$, the memory feature map $\phi(q) = \text{Linear}_\theta(q) \in \mathbb{R}^{d_{\text{mem}}}$ is used for reading/writing the memory. Unless stated otherwise, all attention softmax scalings use $\sqrt{d_{\text{model}}}$.

## E. PyTorch like pseudo code.

---

**Algorithm** PyTorch-like Indexer and Memory module

---

```
# Inputs:
# X: [L, d_model] hidden states
# Q: [H, L, d_head] pre-RoPE query states
# Mask: [L, L] causal mask (added to logits)
# Q_set: query indices for aggregation (prefill: all; decode: window)
# cache: k_cache grows during decoding
def indexer(X, Q, Mask=None, Q_set=None, use_cache=False):
    # down-project multi-head queries and normalize
    q = U_q(flatten(Q, dims=(0,1))) # [L, H_index *d_index]
    q = q.view(L, H_index, d_index)
    q = norm(q) # QK-Norm

    # shared key for all heads (MQA-style)
    k = U_k(X) # [L, d_index]
    k = norm(k)

    # optional cache reuse in decoding
    if use_cache:
        k_cache = cat(k_cache, k, dim=0)
        K = k_cache
    else:
        K = k

    # head gate from hidden state
    alpha = G(X) / sqrt(H_index *d_index) # [L, H_index]

    # gated similarity + causal mask
    z = act(einsum("lhd,td->lth", q, K)) # [L, L, H_index]
    A = sum_h(alpha[:, h] *z[:, :, h]) # [L, L]
    if Mask is not None:
        A = A + Mask

    # per-token importance via max aggregation
    if Q_set is None: Q_set = range(L)
    imp = max(A[Q_set, :], dim=0) # [L]
    return A, imp

# Latent memory (shared across heads)
# State: M [d_mem, d_model], b [d_mem]
def mem_read(q):
    phi_q = Linear_theta(q) # [d_mem]
    denom = (phi_q**2 @ b) + eps
    return (phi_q.T @ M) / denom # [d_model]

def mem_write(evicted_k, evicted_v):
    # evicted_v is summed across heads per token
    phi_k = Linear_theta(evicted_k) # [N, d_mem]
    M = lambda *M + eta *sum(outer(phi_k[i], evicted_v[i]) for i in range(N))
    b = lambda *b + eta *sum(phi_k[i]**2 for i in range(N))

# Residual compensation
o = attn(q, KV_kept) + g(q) *mem_read(q)
```

---

**Algorithm**  Streaming KL Distillation Loss for Indexer Training

```python
# Inputs:
# Q_t, K_t: teacher query/key states, [L, d_head]
# X: hidden states used by the indexer, [L, d_model]
# sink_mask: bool mask on key axis, True for sink tokens, [L]
# Q_set: query indices used for aggregation
# Output:
# loss: KL(softmax(max_q T) || softmax(max_q A))

def streaming_indexer_kl_loss(
    Q_t, K_t, X, indexer, sink_mask=None,
    Q_set=None, q_blk=128, k_blk=4096, causal=True,
):
    L, device = K_t.shape[0], K_t.device
    scale = 1.0 / sqrt(Q_t.shape[-1])

    if Q_set is None: Q_set = arange(L, device=device)

    teacher_imp = full((L,), -inf, device=device, dtype=float32)
    student_imp = full((L,), -inf, device=device, dtype=float32)

    for qb in range(0, len(Q_set), q_blk):
        q_ids = Q_set[qb : qb + q_blk] # [Bq]

        q_t = Q_t[q_ids].float() # [Bq, d_head]

        q_i = indexer.query_features(X, q_ids) # indexer-specific shape

        for kb in range(0, L, k_blk):
            k_ids = arange(kb, min(kb + k_blk, L), device=device)

            k_t = K_t[k_ids].float() # [Bk, d_head]

            T_block = matmul(q_t, k_t.T) *scale # [Bq, Bk]

            A_block = indexer.score_block(X=X, q_ids=q_ids, k_ids=k_ids, q_feat=q_i) # [Bq,
                Bk]

            if causal:
                invalid = k_ids[None, :] > q_ids[:, None]
                T_block = T_block.masked_fill(invalid, -inf)
                A_block = A_block.masked_fill(invalid, -inf)

            teacher_blk = T_block.max(dim=0).values # [Bk]
            student_blk = A_block.max(dim=0).values # [Bk]

            teacher_imp[k_ids] = maximum(teacher_imp[k_ids], teacher_blk)
            student_imp[k_ids] = maximum(student_imp[k_ids], student_blk)

    valid = ones((L,), dtype=bool, device=device)
    if sink_mask is not None:
        valid = valid & (~sink_mask)

    teacher_imp = teacher_imp[valid]
    student_imp = student_imp[valid]

    teacher_logp = log_softmax(teacher_imp, dim=-1)
    student_logp = log_softmax(student_imp, dim=-1)

    teacher_prob = teacher_logp.exp()
    loss = kl_div(input=student_logp, target=teacher_prob, reduction="sum")

    return loss
```

