# OpenReview forum: "IndexMem: Learned KV-Cache Eviction with Latent Memory for Long-Context LLM Inference"
_ICML.cc/2026/Conference — ICML 2026 regular_

### Official Review · Reviewer_9QXL · 2026-03-10

**Soundness:** 3
**Presentation:** 3
**Significance:** 3
**Originality:** 3
**Overall Recommendation:** 4
**Confidence:** 4

**Summary:**

This paper proposes IndexMem, a learnable KV cache eviction framework for long-context LLM inference. It introduces two complementary components: (1) a lightweight MQA-style indexer that predicts token importance by distilling the backbone's attention distribution, enabling more accurate eviction decisions than existing heuristics; and (2) a latent memory module that compresses evicted tokens into a fixed-size state via fast-weight outer-product accumulation, providing gated residual readouts to compensate for information lost through eviction. The backbone LLM is frozen during training; only the indexer (~20M params) and memory (~0.5M params) are optimized. Experiments on Qwen3-8B, Mistral-7B, and Llama-3.1-8B demonstrate consistent improvements on RULER (4K/16K), Needle-in-a-Haystack, and LongBench across a wide range of compression ratios, with particularly large gains (up to +25 points) under aggressive eviction.

**Compliance With Llm Reviewing Policy:**

Affirmed.

**Final Justification:**

I thank the authors for the helpful rebuttal. The responses partially addressed my concerns and improved my understanding of the work. However, some key issues remain, so I will keep my overall evaluation unchanged.

**Key Questions For Authors:**

1.How sensitive are the results to the choice of training data? Have you experimented with other long-context SFT datasets beyond LongAlpaca?
2.What are the actual values of λ and η used in the experiments? How does performance change if λ is set closer to 0 (aggressive forgetting) or closer to 1 (near-perfect retention)?
3.Have you implemented and benchmarked the pre-eviction mode described in Section 3.1? If so, what is the actual prefill speedup?
4.How does the memory readout quality degrade as context length increases significantly beyond the training distribution (e.g., 64K or 128K tokens)?
5.Could the indexer be trained end-to-end with a language modeling loss (rather than KL distillation from backbone attention), and would this improve eviction decisions?

**Limitations:**

YES

**Strengths And Weaknesses:**

**Strengths:**
- The decomposition into "what to evict" (indexer) and "how to recover evicted information" (memory) addresses two genuinely distinct failure modes of KV cache eviction. The indexer improves precision of eviction decisions, while the memory module provides a safety net for the inevitable errors. The NIAH heatmaps (Figure 3) offer compelling visual evidence: the indexer alone achieves high retrieval quality but exhibits rare catastrophic misses, and the memory module effectively eliminates these isolated failures.
- The decision to add memory readout as an explicit residual (Eq. 4) rather than injecting latent tokens into softmax is well-argued. The authors correctly identify the pitfalls of memory-as-tokens (sink collapse, positional constraints, softmax amplification of mismatch) and propose a cleaner alternative. The slow/fast weight decomposition is theoretically grounded in the associative memory literature and practically effective.
- The SnapKV+Memory experiment demonstrates that the latent memory is not tightly coupled to the learned indexer and can serve as a drop-in component for other eviction policies. This substantially broadens the practical impact of the contribution and suggests the memory module is a general-purpose tool for mitigating eviction-induced forgetting.
- The paper evaluates across three model families, three benchmarks (RULER at two context lengths, NIAH, LongBench), and a wide sweep of compression ratios (10%–90%). All methods are implemented within the same KVPRESS framework with identical eviction protocols, which strengthens the credibility of comparisons. The inclusion of both retrieval-focused (RULER, NIAH) and holistic reasoning (LongBench) benchmarks provides a well-rounded assessment.
- The streaming FlashAttention-style computation of max-pooled KL divergence avoids materializing the full L×L attention matrix, making the training objective tractable for long sequences. The two-stage training protocol (indexer first, then joint indexer+memory) is a sensible curriculum that prevents the memory from masking indexer deficiencies.
**Weaknesses**
- IndexMem benefits from supervised distillation on LongAlpaca, while all baselines (SnapKV, PyramidKV, TOVA, KeyDiff, ExpectedAttention) are training-free heuristics. This makes it difficult to disentangle the benefit of the specific architectural design from the benefit of having any learned scoring function. The authors should acknowledge this asymmetry explicitly and, ideally, include at least one learned baseline (e.g., a simple MLP trained with the same distillation objective) to isolate the architectural contribution.
- All experiments use 7B–8B parameter models. The paper motivates the problem with a 70B/1M-token example (320GB KV cache), but does not validate at this scale. Attention patterns, head specialization, and the effectiveness of distillation-based importance estimation may differ substantially in deeper, wider models. Even a single experiment on a 30B+ model would strengthen the claims.
- The latent memory has fixed size M ∈ R^{d_mem × d_model} with d_mem = d/8. Under the decay factor λ, older evicted information fades progressively. The paper does not analyze how memory readout quality degrades as the number of eviction-write cycles grows. For very long sequences (e.g., 128K+), the memory would undergo many write-decay cycles, and it is unclear whether meaningful information from early evictions is retained. An analysis of retrieval accuracy as a function of distance (in eviction cycles) from the original eviction would be informative.

---

> ### Author Rebuttal · Authors · 2026-03-31
>
> **Q1. Comparison with learnable methods**
>
> **A1.**
>
> We agree and evaluated **Locret**, a learned retention method on Llama3-8B-Instruct 75% compression on longbench.
>
> |          | wikiqa | hotpotqa | triviaqa | passage_retrieval_en | multifieldqa_en | multi news | multifieldqa_zh |
> | - | - | - | - | -| - | - | - |
> | IndexMem | 50.97  | 68.67 | 91.00 | 40.00 | 59.80 | 25.40 | 56.58 |
> | Locret [1]   | 10.10  | 13.20 | 59.55 | 85.07 | 16.94 | 19.13 | 16.51 |
> ---
> [1] Locret: Enhancing Eviction in Long-Context LLM Inference with Trained Retaining Heads on Consumer-Grade Devices
>
> **Q2. Experiment on larger model**
>
> **A2.**
>
> We agree that a 30B+ experiment would strengthen the paper. Unfortunately, due to the limited rebuttal timeline, we were not able to add new large-model training. In our current setup, even training the **8K indexer** on a **70B** backbone is already a substantial systems challenge and would likely require **context parallelism (CP)**. We agree this is an important direction and will prioritize it in future work.
>
> ---
>
> **Q3. Memory readout**
>
> **A3.**
>
> Our current memory design uses a relatively simple **additive fast-weight update**. It does not yet incorporate more advanced mechanisms such as **delta-rule updates**[2] or **gated memory writes**[3], which may make it more vulnerable to degradation under repeated write cycles. We view this as an important limitation of the current design, and plan to explore more expressive memory-update mechanisms in future work.
>
> [2] Parallelizing Linear Transformers with the Delta Rule over Sequence Length
>
> [3] Gated Delta Networks: Improving Mamba2 with Delta Rule
>
> ---
>
> **Q4. Other long-context SFT datasets**
>
> **A4.**
>
> We extended training data beyond LongAlpaca and additionally used **nvidia/OpenMathReasoning** and **SWE-bench/SWE-smith-trajectories**, which introduce substantially different reasoning and code/agentic trajectories.
>
> And also updated the training schedule to a more stable **WSD** schedule: **100 warmup steps to $1\times10^{-3}$**, **2000 stable steps**, and 2000 decay steps to $7.5\times10^{-6} $.
>
> Separately, these runs also reinforce an observation consistent with **IndexCache**[4] and our appendix: important token sets show strong overlap across neighboring layers. Motivated by this, we adopt the IndexCache-style score reuse strategy in later experiments, which further reduces indexer overhead without changing the core token-selection objective.
>
> | Method                  | 4k-25% | 4k-50% | 4k-90% | 16k-25% | 16k-50% | 16k-90% |
> | - | :- | :- | :- | :- | :--- | :----- |
> | IndexMem-IndexCache-new | 92.4   | 90.0   | 76.1   | 82.5    | 79.3 | 66.8 |
> | AdaKV                   | 72.8   | 56.7   | 22.8   | 68.6  | 52.1 | 23.8 |
> | SnapKV                  | 57.6   | 41.3   | 19.8   | 49.5 | 36.5 | 21.3 |
>
> ---
>
> **Q5. λ and η**
>
> **A5.**
>
> In our implementation, $\lambda$ and $\eta$ are not fixed global hyperparameters. Instead, we parameterize them as learnable per-layer scalars, with $\lambda$ constrained to $(0,1)$ and $\eta>0$. We initialize them conservatively at $\lambda=0.99$ and $\eta=10^{-3}$, and optimize them jointly with the memory module. This design is motivated by the fact that different layers may require different memory time scales and write strengths.
>
> ---
>
> **Q6. Prefill speedup?**
>
> **A6.**
>
> We have implemented and benchmarked the **pre-eviction** mode.
>
> And consistent with the observations in the appendix of **IndexCache** [4], important token indices exhibit strong overlap across neighboring layers. Motivated by this, we further evaluated an **index-cache** variant that reuses indexer scores for every four consecutive layers, which further reduces the score-computation overhead.
>
> The measured **prefill throughput** on **H2O** with **Llama3-8B** is as follows:
>
> |                                   | bsz1-16K prefill | bsz1-32K prefill | bsz1-128K prefill |
> | --------------------------------- | ---------------- | ---------------- | ----------------- |
> | Baseline                          | 8142 token/s     | 7772 token/s     | 5605 token/s      |
> | IndexMem-pre-eviction             | 8214 token/s     | 8061 token/s     | 8788 token/s      |
> | IndexMem-pre-eviction-index-cache | 8293 token/s     | 8133 token/s     | 8869 token/s      |
>
> [4] IndexCache: Accelerating Sparse Attention via Cross-Layer Index Reuse
>
> ---
>
> **Q7. End to end training with lm loss**
>
> **A7.**
>
> This is a very important question. In our current **score-driven top-$K$** KV-eviction framework, the hard top-$K$ selection is discrete, so LM gradients cannot be directly backpropagated through the token-selection step. Therefore, we train the indexer with a proxy objective, KL distillation from backbone attention, instead of pure LM loss.
>
> To make the selector fully end-to-end trainable, one would likely need either **STE-based gradient approximation** or a **reformulated MoE-like sparse attention mechanism**, where the scores directly modulate the attention outputs.

---

> > ### Author Rebuttal · Reviewer_9QXL · 2026-04-02
> >
> > Thank you for your reply. I will maintain my score.

---

> > > ### Author Response · Authors · 2026-04-04
> > >
> > > Dear Reviewer 9QXL,
> > >
> > > We are glad that we have addressed all of your concerns. Thank you for your response!
> > >
> > > Best regards,
> > >
> > > Authors

---

### Official Review · Reviewer_fTiX · 2026-03-11

**Soundness:** 3
**Presentation:** 3
**Significance:** 3
**Originality:** 3
**Overall Recommendation:** 4
**Confidence:** 4

**Summary:**

This paper studies KV-cache eviction for long-context LLM inference. The core premise is that existing eviction methods are mostly heuristic, so they can mis-estimate which cached tokens will matter later, and once tokens are evicted, their information is permanently lost. To address these two issues, the paper proposes IndexMem, which combines: (i) a learnable indexer that predicts per-token importance for KV retention by distilling from the backbone attention distribution, and (ii) a latent memory module that writes evicted token information into a compact online-updated state and reads it back as a residual compensation term added to the compressed-attention output. The method is trained with the backbone frozen, first training the indexer and then jointly training the indexer and memory module. Empirically, the paper reports improved robustness over prior eviction baselines on RULER, Needle-in-a-Haystack, LongBench, and decoding-time compression experiments across Qwen3-8B, Mistral-7B, and Llama-3.1-8B.

**Compliance With Llm Reviewing Policy:**

Affirmed.

**Final Justification:**

My final recommendation remains Weak Accept, and I am not changing my score. I continue to view the paper as technically sound, clearly presented, and practically meaningful, especially because the gains in the high-compression regime are strong and consistent across models. The rebuttal was helpful: it clarified that the main novelty is the use of memory compensation in the KV-eviction setting rather than the memory mechanism alone, added useful details on training/inference cost, and partially addressed my evaluation-scope concerns with additional baseline and larger-scale evidence. That said, my main reservations are only partially resolved: the memory component still appears only moderately novel relative to prior fast-weight/linear-attention style memory ideas, and the empirical scope still does not fully match the paper’s very long-context motivation. Overall, I think this is a solid and useful paper with clear strengths and some limitations in originality and evaluation breadth, which is why I am keeping my original Weak Accept assessment.

**Key Questions For Authors:**

1 - How does the learned indexer generalize to domains far from LongAlpaca, specifically code, multilingual text, and structured data (e.g., tables, JSON)? Even a small transfer experiment would address a major concern.

2 - Can you provide a precise technical comparison between your memory module's update/readout rule and Titans' memory mechanism? Specifically, what is the architectural delta, and do you have ablations showing your design choices (e.g., the normalization in Eq. 5, the shared-across-heads design) are superior to Titans' formulation? This would address the originality concern.


3 - What is the total training cost (GPU-hours, dataset tokens) for both stages? And what is the prefill-time overhead of running the indexer on, say, a 32K or 128K prompt? The current efficiency analysis only covers decoding.

**Limitations:**

Yes.

**Strengths And Weaknesses:**

## Strengths

1 - The paper is technically coherent at a high level. The method is well motivated by two plausible failure modes of prior work: heuristic token scoring and irreversible forgetting after eviction.

2 - The paper is generally easy to follow. The problem setup is clear, the unified formulation of KV eviction is useful, and the architecture figures help communicate the split between the main attention path and the memory residual path.

3 - The gains at 75–90% compression are substantial (up to +25 points on RULER) and consistent across three model families. This is the regime that matters most practically, and the paper delivers here.

---

## Weaknesses

1 - The memory module's novelty is limited; the connection to linear attention / fast-weight memories is underexplored. The outer-product accumulation with decay (Equations 6–7) and the normalized readout (Equation 5) are essentially a variant of linear attention with a delta-rule-like write, closely related to Titans [1], RWKV-style linear attention, and the fast-weight programmer literature [2]. There are also some missing important baselines and comparisons: 1) H2O (Zhang et al., 2023) is discussed in related work but never compared against experimentally, despite being one of the most cited KV eviction methods. 2) No comparison against learned eviction methods; the paper positions itself as "learnable vs. heuristic," but doesn't compare against TOVA [3] (which is also trained), or other learned cache management methods like AdaKV[4].

2 - Training is done on LongAlpaca only, which is a relatively small and narrow dataset. There is no evaluation of how well the learned indexer generalizes across domains significantly different from the training distribution, e.g., code, multilingual text, or structured data. The paper acknowledges "limited token budgets and modest-scale settings" as a limitation, but this is a significant concern for a method whose core selling point is learned importance prediction. If the indexer overfits to LongAlpaca's distribution, the gains on RULER/LongBench may not transfer.


3 - All experiments use 7–8B models. I think at least one experiment on a larger model (e.g., 70B) or a longer context (32K+, 128K) would be desirable. RULER-16K is the longest evaluation, but the paper motivates the work with 1M-token contexts. The gap between motivation and evaluation is large.



[1] Behrouz, Ali, Peilin Zhong, and Vahab Mirrokni. "Titans: Learning to memorize at test time." arXiv preprint arXiv:2501.00663 (2024).

[2] Schlag, Imanol, Kazuki Irie, and Jürgen Schmidhuber. "Linear transformers are secretly fast weight programmers." International conference on machine learning. PMLR, 2021.

[3] Oren, Matanel, et al. "Transformers are multi-state rnns." Proceedings of the 2024 Conference on Empirical Methods in Natural Language Processing. 2024.

[4] Feng, Yuan, et al. "Ada-kv: Optimizing kv cache eviction by adaptive budget allocation for efficient llm inference." arXiv preprint arXiv:2407.11550 (2024).

---

> ### Author Rebuttal · Authors · 2026-03-30
>
> **Q1. connection to prior fast-weight memory literature & comparison with Titan**
>
> **A1.**
>
> We appreciate this point. We do **not** claim the memory module itself as the main novelty of the paper. Our contribution is to **introduce memory into the KV-eviction setting** to mitigate the irreversible forgetting caused by token eviction.
>
> Our current design is intentionally simple: it uses an **additive fast-weight update**, rather than a delta-rule[1] write. We agree that its connection to prior linear-attention / fast-weight memory literature should be discussed more explicitly, including fast-weight programmers, RWKV-style linear attention, and Titans.
>
> We view this memory module as a **minimal instantiation**, not the final design point. A natural next step is to explore stronger updates such as **delta-rule**[1] and **gated** memory writes[2].
>
> We also note that Titans is closer to a **sliding-window attention + memory** design, while our work studies **token eviction + memory compensation**.
>
> In addition, memory-as-latent-token designs such as **MSA**[3] are also promising extensions of our framework.
>
>
>
> [1] Parallelizing Linear Transformers with the Delta Rule over Sequence Length
>
> [2] Gated Delta Networks: Improving Mamba2 with Delta Rule
>
> [3] MSA: Memory Sparse Attention for Efficient End-to-End Memory Model Scaling to 100M Tokens
>
> ---
>
> **Q2. Limited training dataset of LongAlpaca**
>
> **A2.**
>
> We agree that training only on **LongAlpaca** is limited for a learned importance predictor. To reduce the risk of overfitting to a narrow distribution, we extended training data beyond LongAlpaca and additionally used **nvidia/OpenMathReasoning**[4] and **SWE-bench/SWE-smith-trajectories**[5], which introduce substantially different reasoning and code/agentic trajectories.
>
> We also updated the training schedule to a more stable **WSD** schedule: **100 warmup steps to $1\times10^{-3}$**, **2000 stable steps**, and **2000 decay steps to $7.5\times10^{-6}$**.
>
> Separately, our appendix is also consistent with **IndexCache**[6]: important token sets show overlap across neighboring layers. Motivated by this, we adopt the IndexCache-style score reuse strategy.
>
> [4] https://huggingface.co/datasets/nvidia/OpenMathReasoning
>
> [5] https://huggingface.co/datasets/SWE-bench/SWE-smith-trajectories
>
> [6] IndexCache: Accelerating Sparse Attention via Cross-Layer Index Reuse
>
> ---
>
> **Q3. More baseline comparisons.**
>
> **A3.**
>
> **H2O.** We did not include H2O because it is primarily a **decode-oriented dynamic eviction** method: its heavy-hitter policy relies on attention statistics accumulated over decoding steps, whereas our main setting is **long-prefill compression**, where retention decisions must be made immediately after processing the prompt. Our paper focuses on the **long-prefill, short-decode** regime, and in this setting prompt-side baselines such as **SnapKV** are more directly comparable, since they are designed to predict important prompt tokens from the prefill stage itself.
>
> Additional baselines on Mistral-7B:
>
>
> | Method | 4k-25% | 4k-50% | 4k-90% | 16k-25% | 16k-50% | 16k-90% |
> | ------ | :--- | :--- | :--- | :--- | :--- | :--- |
> | IndexMem-IndexCache-new | 92.4 | 90.0 | 76.1 | 82.5 | 79.3 | 66.8 |
> |  AdaKV | 72.8 | 56.7 | 22.8 | 68.6 | 52.1 | 23.8 |
> | SnapKV | 57.6 | 41.3 | 19.8 | 49.5 | 36.5 | 21.3 |
> | TOVA | 78.4 | 75.5 | 29.2 | 73.8 | 70.1 | 47.7 |
>
> ---
>
> **Q4. Longer evaluation benchmark**
>
> **A4.**
>
> We agree that there is a gap between our long-context motivation and the current evaluation scale. Beyond ruler 16K, our **LongBench** evaluation includes longer real examples, with the longest instances reaching **40K tokens**.
>
> Under our current **8K training setting**, the indexer+memory generalizes well to **32K** contexts, but performance begins to deteriorate when extrapolated to **128K** or **1M** tokens.
>
> Our current training uses **DDP without context parallelism**, which limits the feasible training length. We agree that experiments on larger models and substantially longer contexts would strengthen the paper, and we plan to extend in this direction in future work.
>
> ---
>
> **Q5. Detail time cost in training and inference**
>
> **A5.**
>
> Our total training cost for the two-stage procedure is **4100 steps** with **global batch size 128**, corresponding to **524800 training samples** drawn from our math and SWE datasets. Training was run on **8×H20 GPUs for 52 hours**.
>
> Prefill overhead measured on H20 with Llama3-8B:
>
> |                                   | bsz1-16K prefill | bsz1-32K prefill | bsz1-128K prefill |
> | --------------------------------- | ---------------- | ---------------- | ----------------- |
> | Baseline                          | 8142 token/s     | 7772 token/s     | 5605 token/s      |
> | IndexMem-pre-eviction             | 8214 token/s     | 8061 token/s     | 8788 token/s      |
> | IndexMem-pre-eviction-index-cache | 8293 token/s     | 8133 token/s     | 8869 token/s      |

---

> > ### Author Rebuttal · Reviewer_fTiX · 2026-04-03
> >
> > Thank you for the rebuttal. It resolves a substantial part of my concerns. In particular, the clarification that the memory module itself is not the primary novelty claim, but rather a way to introduce memory into the KV-eviction setting, makes the contribution clearer. The additional discussion distinguishing this from Titans/fast-weight memory work, the added comparison against AdaKV, and the concrete training-cost / prefill-overhead numbers are all helpful.
> >
> >
> > My concerns are therefore partially resolved, but not fully. The main remaining issue is evaluation scope. The rebuttal improves the training-data concern by stating that training was extended beyond LongAlpaca, but it still does not provide a direct transfer evaluation on clearly out-of-domain settings such as multilingual or structured inputs, which was the core of my question. Likewise, the rebuttal acknowledges that the current setting generalizes to 32K better than to 128K/1M, and the paper still does not include a larger-model experiment; so the gap between the long-context motivation and the demonstrated evaluation range remains.

---

> > > ### Author Response · Authors · 2026-04-06
> > >
> > > **Q. Evaluation scope**
> > > **A.**
> > > The original submission already contained evidence relevant to this concern. LongBench v1 evaluates bilingual long-context understanding, with both Chinese and English tasks across multiple categories, thereby providing a multilingual transfer test beyond a purely English setting.
> > > To further respond to the reviewer’s concern about model scale, we additionally trained and evaluated a Llama-3.3-70B model. Moreover, this additional evaluation is conducted on LongBench v2, which explicitly includes a long structured data understanding subset in addition to standard free-form QA-style settings. Therefore, the new experiment extends the model scale, and provides direct evidence on structured-input generalization.
> > > The corresponding results are reported below.
> > > | Method | Overall (%) | Easy (%) | Hard (%) | Short (%) | Medium (%) | Long (%) |
> > > |---|---:|---:|---:|---:|---:|---:|
> > > | Llama-3.3-70B | 29.8 | 34.4 | 27.0 | 36.7 | 27.0 | 24.1 |
> > > | TOVA | 18.0 | 11.1 | 21.9 | 17.6 | 14.3 | 25.0 |
> > > | IndexMem | 28.0 | 16.7 | 34.4 | 29.4 | 23.8 | 33.3 |

---

### Official Review · Reviewer_pcVp · 2026-03-12

**Soundness:** 2
**Presentation:** 2
**Significance:** 2
**Originality:** 2
**Overall Recommendation:** 4
**Confidence:** 5

**Summary:**

The authors use a learned indexer to pick tokens to keep and evict the rest. The evicted tokens are compressed to a memory which helps in recovering evicted information. This can reduce the KV-Cache overhead significantly, and also make the attention cheaper because fewer tokens are retaining full KV-Cache

**Compliance With Llm Reviewing Policy:**

Affirmed.

**Final Justification:**

I was convinced by the authors response and baselines so I raised it to a weak accept.

**Key Questions For Authors:**

- Am I missing any results on scaling the memory and seeing the accuracies change (improve)? A sweep over d_{mem} and even the memory write read parameters would be erally interesting...
- Any further notes on the performance difference being lesser than one would expect out of eviction based sparsity would be appreciated.

**Limitations:**

yes

**Strengths And Weaknesses:**

Strengths:
- This is a very relevant problem, as several research papers look at token sparsity, and eviction methods consistently significantly underperform methods that retain the tokens.
- Empirically, the method is tested on several models and benchmarks. The idea of using a latent state to retain evicted token information is interesting.

Weaknesses:
- A key contribution is a learned indexer, which misses several relevant prior references (TRIM-KV, TokenButler, Learning to Evict from Key-Value Cache) that would contextualize the paper in relevant recent work.
- On RULER, several works that compress the KV-Cache 8x (roughly 90% compression rate, for e.g., xKV Table 1) only see a degradation of 4% in accuracy. How did Table 1 go from 93.4 to 56%? Was the instruction-tuned variant used?
- The LongBench evaluation should ideally be averaged over all tasks in Figure 4.
- Figure 7 IndexMem seems to really get closer to the average decoding time of the baseline, given the compression ratio of 0.9, where the accuracy goes down to 56% on RULER, it seems a bit confusing to trade < 0.1 seconds of decode time for half the long-context accuracy. The performance evaluation is not entirely  convincing and needs work.

---

> ### Author Rebuttal · Authors · 2026-03-31
>
> **Q1. Discuss relevant recent work**
>
> **A1.**
>
> We will cite and discuss TRIM-KV, TokenButler, and Learning to Evict from Key-Value Cache in Section 2.2. They can all be viewed through the lens of our unified formulation.
>
> ---
>
> **Q2. About xKV**
>
>
> **A2.**
>
> In our experiments, we indeed use the instruction-tuned Llama variant.
>
> And we find that this behavior is closely related to the task characteristics. RULER is a highly sparse retrieval-style benchmark, where performance is dominated by whether a very small set of critical evidence tokens is preserved. On such sparse tasks, IndexMem is not always stronger than xKV under extremely aggressive compression. However, on information-dense long-context understanding tasks, where useful evidence is distributed across many tokens, IndexMem shows a clear advantage.
>
> For 75% compression on Longbench
>
> |          | wikiqa | hotpotqa | triviaqa | passage_retrieval_en | multifieldqa_en | multi news | multifieldqa_zh |
> | - | - | - | - | - | - | - | - |
> | IndexMem | 50.97  | 68.67    | 91.00    | 40.00 | 59.80 | 25.40  | 56.58 |
> | xKV      | 46.21  | 55.81    | 90.00    | 1.00 | 32.61 | 25.04 | 46.13 |
> | Locret[1]   | 10.10 | 13.20 | 59.55 | 85.07 | 16.94 | 19.13 | 16.51 |
> ---
> [1] Locret: Enhancing Eviction in Long-Context LLM Inference with Trained Retaining Heads on Consumer-Grade Devices
>
> **Q3. Average over all tasks of Longbench**
>
> **A3.**
>
> Here, 10%, 25%, etc. denote the compression ratio. IndexMem consistently outperforms the competing methods, and we also observe that at certain compression levels, stronger compression can even improve accuracy, likely because eviction increases the information density of the retained cache.
>
> |                    | 10%   | 25%   | 50%   | 75%   | 90%   |
> | ------------------ | ----- | ----- | ----- | ----- | ----- |
> | IndexMem           | 53.48 | 53.18 | 55.26 | 42.18 | 36.51 |
> | Expected Attention | 43.78 | 43.88 | 42.09 | 36.33 | 28.60 |
> | PyramidKV          | 43.63 | 41.57 | 42.35 | 39.17 | 29.68 |
> | SnapKV             | 43.55 | 43.37 | 41.44 | 39.22 | 29.40 |
> | TOVA               | 43.81 | 43.08 | 42.10 | 39.05 | 32.56 |
>
> ---
>
> **Q4. Efficiency of IndexMem**
>
> **A4.**
>
> Thank you for this important question!
>
> The setting in Figure 7 is: **batch size = 1** with **32K prefill and 1K decode**. KV-cache memory decreases from 4GB to 0.8GB. However, since model weights (16GB) remain the primary memory bottleneck, decoding is still weight-bound rather than KV-bound. Consequently, latency gains from KV eviction are limited in this setting. (Cache memory reported in Figure 7 includes both activation and KV cache. We will correct this in the upcoming revision to represent the KV cache exclusively.)
>
> In addition, Fig7 not yet implement the **pre-eviction**(mentioned in line174). Our indexer is suited for predicting which KV entries are important **before** materializing the full cache, so that only the selected KV are computed and stored during prefill, i.e., a sparse-prefill style execution.
>
> We expect this to be beneficial in **longer-context** or **larger-batch** scenarios, where full KV cache creates memory pressure and requires chunked prefill.
>
> Finally, consistent with appendix observations in IndexCache[2], important token indices show strong overlap across neighboring layers. Motivated by this, we also tested an index-cache variant that reuses indexer scores across every four layers.
>
> |                                   | bsz4-P32k -D128 | bsz16-P32k-D128(chunk prefill) | bsz4-P128k-D128(chunk prefill) |
> | -- | -- | -- | -|
> | Baseline                          | 4216 ms         | 16282 ms                       | 23383 ms                       |
> | IndexMem-pre-eviction             | 3964 ms         | 12315 ms                       | 14914 ms                       |
> | IndexMem-pre-eviction-index-cache | 3579 ms         | 10346 ms                       | 11788 ms                       |
>
> All results are measured on **H20**, with **compression ratio = 75%** and **chunk-prefill segment size = 8192**.
>
> [2] IndexCache: Accelerating Sparse Attention via Cross-Layer Index Reuse
>
> ---
>
> **Q5. A sweep over $d_{\text{mem}}$ and the memory write/read parameters**
>
> **A5.**
>
> $d_{\text{mem}}$ is the main architectural hyperparameter in our memory module, while the remaining dimensions are inherited from the backbone hidden size. We use $d_{\text{mem}} = d/8$ in all experiments, as it offers a practical trade-off between capacity and inference cost: increasing $d_{\text{mem}}$ further also increases the projection and memory read/write GEMM overhead during inference.
>
> More broadly, we believe the current online fast-weight update is still relatively simple. Strengthening it with a lightweight gating mechanism[3] and a delta-rule style update[4] could allow the model to make better use of larger $d_{\text{mem}}$.
>
> [3] Gated Delta Networks: Improving Mamba2 with Delta Rule
>
> [4] Parallelizing Linear Transformers with the Delta Rule over Sequence Length

---

> > ### Author Rebuttal · Reviewer_pcVp · 2026-04-03
> >
> > Thank you for your response, I appreciate the candid explanations, and I will raise my score as my concerns have been adequately addressed, to a weak accept. For a strong accept I would expect significantly deeper investigation of performance which is not addressable in a short rebuttal.

---

> > > ### Author Response · Authors · 2026-04-04
> > >
> > > Dear Reviewer pcVp,
> > >
> > > We are glad that we have addressed all of your concerns. Thank you for your response!
> > >
> > > Best regards,
> > >
> > > Authors

---

### Official Review · Reviewer_g1jb · 2026-03-13

**Soundness:** 3
**Presentation:** 3
**Significance:** 2
**Originality:** 3
**Overall Recommendation:** 4
**Confidence:** 5

**Summary:**

The paper studies KV cache eviction for long-context LLM inference. It proposes a learnable indexer to estimate token importance and a latent memory module that compresses evicted tokens to reduce information loss. Experiments demonstrate improved performance on several long-context benchmarks under constrained KV budgets.

**Compliance With Llm Reviewing Policy:**

Affirmed.

**Final Justification:**

Once the authors finished the revised version, i will raise the score to 4

**Key Questions For Authors:**

All are the above ones.

**Limitations:**

yes

**Strengths And Weaknesses:**

s1. This work introduces a learnable KV cache eviction mechanism inspired by learnable sparse attention.

s2. The paper provides relatively detailed experiments to demonstrate the effectiveness of the proposed method.

w1. In Lines 130–151, the notation is quite messy and hard to follow. The meanings of Uq, Uk, G and Qs are unclear, and I could not find clear explanations for these symbols. The subscripts and notation make this section difficult to understand.

w2. In the efficiency evaluation, the paper uses a setting of 32K prefill and 1K decoding. Could the authors provide results under different settings, such as 4K + 8K or 32K + 8K, especially for scenarios with longer decoding?

w3. It would also be helpful to compare with ShadowKV and XKV, which appear to be more efficient baselines than those currently reported.

w4. In the discussion of sparse attention, related work such as SeerAttention and DuoAttention should also be discussed, since they also explore learnable attention mechanisms.

---
Reference

[1] ShadowKV: KV Cache in Shadows for High-Throughput Long-Context LLM Inference. arXiv preprint arXiv:2410.21465, 2024.

[2] xKV: Cross-Layer SVD for KV-Cache Compression. arXiv preprint arXiv:2503.18893, 2025.

[3] SeerAttention: Learning Intrinsic Sparse Attention in Your LLMs. arXiv preprint arXiv:2410.13276, 2024.

[4] DuoAttention: Efficient Long-Context LLM Inference with Retrieval and Streaming Heads. arXiv preprint arXiv:2410.10819, 2024.

---

> ### Author Rebuttal · Authors · 2026-03-30
>
> **Q1. More detail notation.**
>
> **A1.**
>
> **Addressing Subscripts:**
>
> * $s$: Denotes the sequence position of the current query.
> * $t$: Denotes the sequence position of the cached key.
> * $h$: Denotes the indexer's attention head index ($h \in \{1, \dots, H_{\text{index}}\}$).
>
> **1. $U_q$**
>
> $U_q$ is a learnable query-side projection, compresses the backbone model's original multi-head query representation $Q_s$ into the lower-dimensional query space used by indexer. Specifically, it flattens the backbone multi-head query at position $s$, and then applies a linear projection.
>
> **2. $U_k$**
>
> $U_k$ is a learnable projection from hidden states to the shared key space. It maps the hidden state $X_t$ into the shared key space used by the indexer:
>
> $k_t = U_k X_t \in \mathbb{R}^{d_{\text{index}}}$ The paper states that the indexer adopts an MQA-style design, meaning the key is a single shared key rather than one per head. This key feature is shared across all indexer heads.
>
> **3. $G$**
>
> $G$ is a gating projection. It maps the hidden state $X_s$ at the current position to a head-wise gating vector $\alpha_s$, which is used to assign a weight to each indexer head:
> $\alpha_s=\frac{G X_s}{\sqrt{H_{\text{index}} d_{\text{index}}}}\in\mathbb{R}^{H_{\text{index}}}$
>
> **4. $Q_s$**
>
> $Q_s$ represents the backbone model's multi-head query state at position $s$.
>
> **5. $z_{s,t}$**
>
> $z_{s,t}\in\mathbb{R}^{H_{\text{index}}}$
> The $h$-th component of this vector is defined as:
> $z_{s,t,h}=act(q_{s,h}^\top k_t)$
>
> ---
>
> **Q2. Long decoding setting**
>
> **A2.**
>
> For the **32K prefill** setting, chunked prefill is required in our current implementation, so here we report additional results on **4K + 8K decode** and **16K + 8K decode**. We also enable both **pre-eviction** mechanism (mentioned in line174).
>
> Furthermore, consistent with appendix observations in IndexCache[1], important token indices show strong overlap across neighboring layers. Motivated by this, we also tested an index-cache variant that reuses indexer scores across every four layers, further reducing the overhead of score computation.
>
> |                                   |        bsz1-4kprefill-8kdecode |       bsz1-16kprefill-8kdecode |
> | :-------------------------------- | -----------------------------: | -----------------------------: |
> | Baseline                          | P: 8589token/s, D: 23.6token/s | P: 8142token/s, D: 20.7token/s |
> | IndexMem-pre-eviction             | P: 8481token/s, D: 27.1token/s | P: 8214token/s, D: 24.3token/s |
> | IndexMem-pre-eviction-index-cache | P: 8527token/s, D: 28.5token/s | P: 8293token/s, D: 25.6token/s |
>
> All results are measured on **H20** with **Llama3-8B**.
>
> [1] IndexCache: Accelerating Sparse Attention via Cross-Layer Index Reuse
>
> ---
>
> **Q3. About xKV/ShadowKV**
>
> **A3.**
>
> We agree that **ShadowKV** and **xKV** are important related methods. However, they are **not directly comparable baselines for the main question studied in our paper**, because they operate on a different compression axis.
>
> Our work studies **token-level KV eviction**, i.e., reducing the cache along the **sequence dimension** under a fixed token budget, with the central question being **which tokens to keep**. In contrast, **ShadowKV/xKV** mainly reduce the **per-token representation cost** of KV cache, rather than addressing token selection itself. These two directions are largely **orthogonal and potentially composable**.
>
> We also find that the relative behavior depends strongly on the **task characteristics**. On **RULER**, which is a highly sparse retrieval benchmark, IndexMem is not always stronger than xKV under very aggressive compression. However, on **information-dense LongBench tasks**, where useful evidence is distributed across many tokens, IndexMem shows a clearer advantage.
>
> For example, at **75% compression on LongBench**:
>
> |          | wikiqa | hotpotqa | triviaqa | passage_retrieval_en | multifieldqa_en | multi_news | multifieldqa_zh |
> | - | - | - | - | - | - | - | - |
> | IndexMem | 50.97  | 68.67    | 91.00    | 40.00                | 59.80           | 25.40      | 56.58           |
> | xKV      | 46.21  | 55.81    | 90.00    | 1.00                 | 32.61           | 25.04      | 46.13           |
>
> We will clarify this distinction in the revision and position **ShadowKV/xKV as complementary, rather than directly competing, approaches**.
>
>
> ---
>
> **Q4. Discuss SeerAttention/DuoAttention**
>
> **A4.**
>
> We will include SeerAttention and DuoAttention in the revised related-work section. SeerAttention introduces adaptive block-level sparse attention, while DuoAttention leverages head-wise heterogeneity via retrieval/streaming head specialization; both are relevant additions because they explore adaptive attention mechanisms for efficient long-context inference.

---

> > ### Author Rebuttal · Reviewer_g1jb · 2026-04-04
> >
> > Thank you for your response. I will raise my score into 4 after you finish the revised version

---

> > > ### Author Response · Authors · 2026-04-05
> > >
> > > Dear Reviewer g1jb,
> > >
> > > We are glad that we have addressed all of your concerns. Thank you for your response!
> > >
> > > Best regards,
> > >
> > > Authors

---

### Decision · Program_Chairs · 2026-04-30

**Decision:**

Accept (regular)

**Comment:**

The paper proposes a scheme for KV-cache eviction in long-context LLMs inference by proposing a learnable indexer to estimate token importance. This is supplemented by a latent memory module to which the evited tokens are compressed to minimize information loss. The reviewers agree that : (i) the paper attempts to solve an important practical and contemporary research challenge faced in LLMs, (ii) The approach itself is well-argued with the ‘why’, ‘what’ and ‘how’ questions answered systematically, (iii) The results in the paper are strong and compelling, experiments are quite comprehensive with improvements shown across three model families, and (iv) The paper is clear in terms of writing and flows logically.

Based on the unanimous positive feedback by the reviewers, the paper is recommended to be accepted.